# Transposable elements as tissue-specific enhancers in cancers of endodermal lineage

Konsta Karttunen[1,5], Divyesh Patel [1,2,5], Jihan Xia[1,2], Liangru Fei[1], Kimmo Palin [1,2], Lauri Aaltonen [1,2] & Biswajyoti Sahu [1,2,3,4] ✉

Transposable elements (TE) are repetitive genomic elements that harbor binding sites for human transcription factors (TF). A regulatory role for TEs has been suggested in embryonal development and diseases such as cancer but systematic investigation of their functions has been limited by their wide-spread silencing in the genome. Here, we utilize unbiased massively parallel reporter assay data using a whole human genome library to identify TEs with functional enhancer activity in two human cancer types of endodermal lineage, colorectal and liver cancers. We show that the identified TE enhancers are characterized by genomic features associated with active enhancers, such as epigenetic marks and TF binding. Importantly, we identify distinct TE sub-families that function as tissue-specific enhancers, namely MER11- and LTR12-elements in colon and liver cancers, respectively. These elements are bound by distinct TFs in each cell type, and they have predicted associations to differentially expressed genes. In conclusion, these data demonstrate how different cancer types can utilize distinct TEs as tissue-specific enhancers, paving the way for comprehensive understanding of the role of TEs as bona fide enhancers in the cancer genomes.

Around half of the human genome consists of sequences that originate from TE insertions. The advent of whole genome sequencing technologies has revealed the major contribution of TEs to the evolution, size, and regulatory functions of eukaryotic genomes[1]. TEs harbor *cis*-regulatory sequences for human TFs and can thus contribute to the human gene regulatory elements such as promoters and enhancers[2,3]. TEs have been shown to have regulatory functions during development[4], and their role in complex genetic diseases such as cancer is also becoming more evident[5]. However, to what extent TEs contribute to gene regulatory functions in different cancer types is still poorly understood.

Most of the TEs in the human genome have been immobilized due to truncations and mutations that accumulate during evolution and the majority of TEs remain epigenetically silenced in normal somatic cells. However, there are ~100 copies of long interspersed nuclear element 1 (LINE-1; L1) that are capable of retrotransposition, i.e.,

inserting themselves into new genomic loci[6]. Accumulating evidence from whole genome sequencing studies indicates that the L1 elements can be widely activated in cancer[7–10] and different cancer types show distinct rates for somatic retrotransposition[11,12]. Aberrant TE insertions can contribute to tumorigenesis by inducing genomic rearrangements that can lead to deletion of tumor-suppressor genes or amplification of oncogenes[12]. Importantly, the retrotransposition-incapable TEs can also play a role in tumorigenic processes by providing a large repository of potential regulatory elements that can be repurposed for transcriptional control of human genes[13]. However, since the activation of the retrotransposition-incapable TEs can occur for example through destabilization of their epigenetic silencing, their activation cannot be detected in whole genome sequencing data and thus their functional significance in cancer cells has remained largely elusive. Few studies have reported activation of proto-oncogenes by derepressed long terminal repeat (LTR)-elements at their promoters[14–16], but systematic

[1]Applied Tumor Genomics Program, Research Programs Unit, Faculty of Medicine, University of Helsinki, Helsinki, Finland. [2]iCAN Digital Precision Cancer Medicine Flagship, University of Helsinki, Helsinki, Finland. [3]Medicum, Faculty of Medicine, University of Helsinki, Helsinki, Finland. [4]Centre for Molecular Medicine Norway, University of Oslo, Oslo, Norway. [5]These author contributed equally: Konsta Karttunen, Divyesh Patel. ✉e-mail: biswajyoti.sahu@helsinki.fi

functional studies and epigenetic profiling are necessary for comprehensive understanding of the regulatory activity of TEs in the cancer genomes.

Here, we leverage on the unbiased and genome-wide episomal enhancer activity measurements from an ultracomplex whole human genome library together with genome-wide epigenetic profiling to characterize TE-derived enhancers in two distinct cancer types, colorectal and hepatocellular carcinomas. We show that the functional activity is similar for a subset of TEs but, importantly, there are distinct TE subfamilies that are activated by different TFs in a highly tissue-specific manner, suggesting that these TEs can act as tissue-specific enhancers resulting in differential gene regulatory activity in human cancers.

## Results

### Long terminal repeats are enriched within active enhancers identified by STARR-seq

To measure the functional enhancer activity of TEs in two endodermal origin cancers, namely colon and liver cancers, we utilized our publicly available massively parallel reporter assay data from the genomic STARR-seq experiments in GP5d and HepG2 cell lines, respectively[17]. We then characterized the enhancer properties of the TEs by combining STARR-seq with epigenetic and regulatory data from the same cell lines, such as ATAC-seq for chromatin accessibility, ChIP-seq for epigenetic marks and TF binding, and nuclear run-on data for detecting the nascent transcripts (Fig. 1a). In the STARR-seq datasets utilized

here, the whole human genomic DNA was cloned into the 3′-UTR of the reporter gene driven by a weak minimal promoter to measure enhancer activity with a ~1.5 bp resolution[17]. We mapped the data to human genome (hg38) and identified the active enhancers by peak calling against plasmid input, resulting in 15,390 peaks for GP5d and 11,951 peaks for HepG2 (Supplementary Data 1 and 2). Over half of the active enhancer peaks in both GP5d and HepG2 overlap with at least one TE (Fig. 1b). As a control, we performed STARR-seq experiment from a non-malignant ectodermal-origin retinal pigment epithelial cell line, RPE1, resulting in 6476 called peaks against input plasmid library (Supplementary Data 3). Analysis of different TE classes [LINE, short interspersed nuclear elements (SINE), LTR, and DNA elements] revealed that the LTR elements were significantly overrepresented within the active enhancer peaks in GP5d, HepG2 and RPE1 cell lines ($p < 2.2 \times 10^{-16}$; Fig. 1c, Supplementary Fig. 1a). In all three cell lines, many LTR subfamilies were highly enriched within STARR-seq peaks with up to 40% of their genomic copies overlapping a peak summit (Supplementary Fig. 1b). However, different TE subfamilies were not equally enriched between the three cell lines (Supplementary Fig. 1b), suggesting cell type-specificity in their activity. Previously, it has been shown that evolutionarily young LTRs are enriched within open chromatin more frequently compared to older LTRs, suggesting that younger LTRs are more active in the genome[18]. This was shown to be independent of their sequence features, such as mappability of TE subfamilies[18]. In agreement with this, we found primate-specific TEs being overrepresented within the active enhancers in all three cell lines

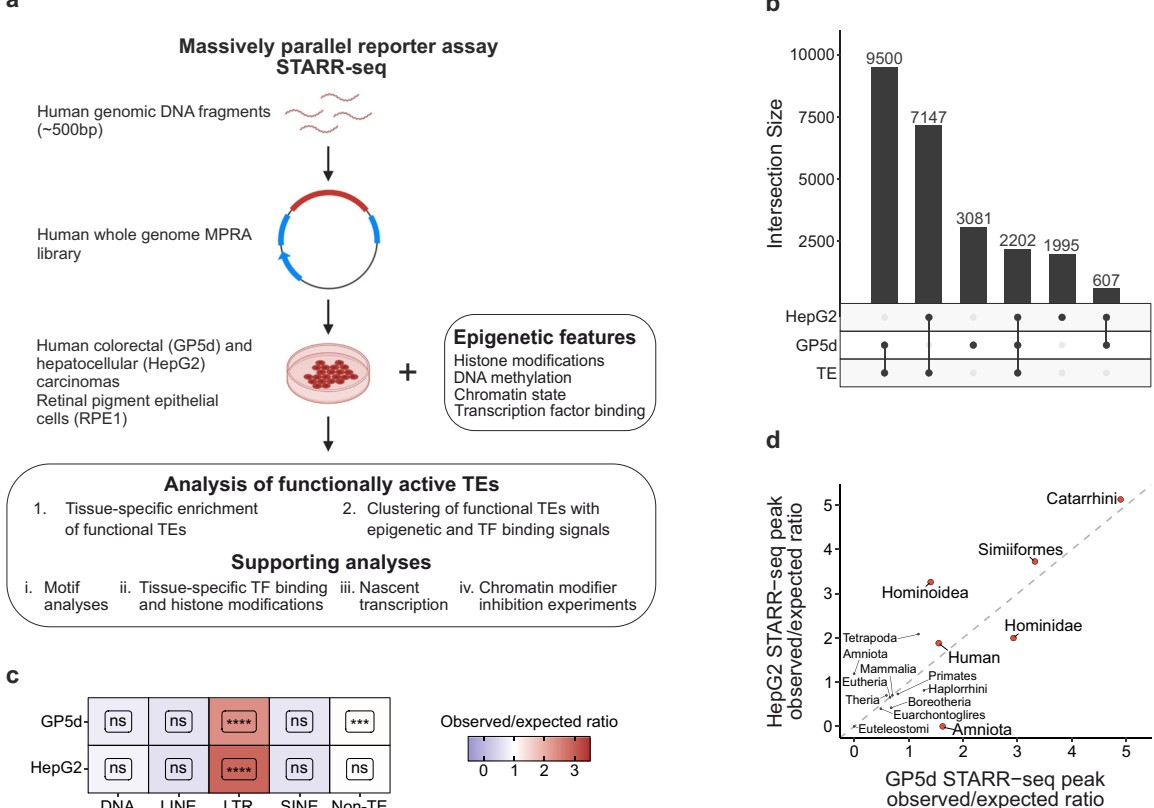

**Fig. 1 | Long terminal repeats are enriched within active enhancers identified by STARR-seq. a** Schematic representation of the analysis pipeline. **b** Upset plot for overlap analysis of STARR-seq peaks with TEs in GP5d and HepG2 cells with the number of peaks in each category indicated; total number of peaks 15,390 and 11,951 in GP5d and HepG2 cells, respectively. **c** Ratio of observed vs. expected overlaps for all GP5d and HepG2 STARR-seq peak summits with the major classes of TEs (DNA, LINE, LTR, and SINE) and the non-TE genome. BH-adjusted one-sided binomial test FDR is shown for each class (Significance symbols: **** indicates $p < 0.0001$, *** $p < 0.001$, ** $p < 0.01$, * $p < 0.05$, ns = non-significant, $p > 0.05$). GP5d LTR $p < 2.2\text{e-16}$, GP5d Non-TE $p = 4.331607\text{e-04}$, HepG2 LTR $< 2.2\text{e-16}$. **d** Enrichment of all STARR-seq peak summits at TEs classified by lineage significant in both GP5d and HepG2 cells. TE subfamilies were grouped by their lineage of origin and the observed/expected ratio of STARR-seq peak summits at the TE lineage groups was calculated. TE lineages significant in both GP5d and HepG2 (BH-adjusted one-sided binomial test FDR < 0.01) are labeled in red, gray points are statistically insignificant. Source data are provided as a Source Data file.

(Fig. 1d, Supplementary Fig. 1c). Taken together, our results show that we can detect enhancer activity from the LTR elements in both colon and liver cancer cells as well as in non-malignant retinal epithelial cells in the STARR-seq assay.

## TEs are enriched for distinct epigenetic signatures and TF binding motifs

To characterize the properties of active enhancers in GP5d cells, unsupervised k-means clustering was performed for all genomic STARR-seq peaks along with the signal for open chromatin from ATAC-seq, ChIP-seq for histone modifications (H3K4me1, H3K9me3, H3K27ac, H3K27me3, and H3K36me3) and p53 binding (with and without 5-fluorouracil treatment) as well as CpG methylation called from long-read nanopore sequencing. Five clusters resulted in optimal clustering (Supplementary Fig. 2a), and each cluster showed distinct enrichment for genomic regulatory features (Fig. 2a). GP5d clusters 1 and 2 with 2367 and 897 peaks harbor the classical features of enhancers and promoters, respectively. Both show strong chromatin accessibility and low levels of CpG methylation, H3K9me3 and H3K27me3 signals (Fig. 2a). Peaks in cluster 1 are enriched for H3K4me1 and H3K27ac histone modifications with a bimodal signal around the peak center that is characteristic to active enhancers, whereas peaks in cluster 2 show very high enrichment for promoter-specific H3K4me3 and less prominent central distribution of H3K27ac signal. In addition, the flanks of the cluster 2 peaks are enriched for H3K36me3 marking the gene bodies, and peak annotation confirmed their association to promoters in the human genome (Supplementary Fig. 2b). Enrichment analysis for TF binding motifs revealed basic leucine zipper (bZIP) family motifs such as JUN/FOS within the peaks in cluster 1 (Fig. 2b), supporting their canonical enhancer function, whereas cluster 2 was enriched for KLF/SP motifs from the bZIP family as well as zinc finger factors from the YY family (Fig. 2b), which are known to be promoter-specific factors[19]. Cluster 2 peaks had very little overlap with TEs (Supplementary Fig. 2c), consistent with previous findings of TE depletion in promoter regions[20]. Interestingly, however, two-thirds of the peaks in cluster 1 overlap with TEs (Supplementary Fig. 2c), speaking for their potential as functional enhancers from the TEs in the endogenous genomic context.

GP5d cluster 3 with 10,741 peaks showed low STARR-seq signal and little enrichment for open chromatin or active histone marks, with around half of the peaks overlapping a TE (Fig. 2a, Supplementary Fig. 2c), suggesting that this cluster mostly comprises peaks that are repressed in the endogenous chromatin context but show enhancer activity in the episomal STARR-seq assay and resemble cryptic enhancers described previously[17]. Motif analysis revealed moderate enrichment for p53 domain as well as zinc finger factor (ZNF) motifs that are mostly absent from other clusters (Fig. 2b), suggesting that repression of the TEs in this cluster is possibly mediated in vivo via known TE suppressors, p53[21] and Krüppel-associated box domain zinc fingers (KRAB-ZNFs)[22]. However, the strongest enrichment for p53 domain motif was observed within clusters 4 ($n = 242$) and 5 ($n = 1143$) (Fig. 2b). These clusters also showed the strongest STARR-seq signal indicating strong enhancer activity (Fig. 2a), and the majority of the peaks were associated with a TE (Supplementary Fig. 2c). The enrichment of active histone marks at the genomic loci corresponding to clusters 4 and 5 was relatively low, but both showed strong p53 binding (Fig. 2a). Interestingly, cluster 4 also showed relatively high enrichment for H3K9me3 signal (Fig. 2a), a repressive epigenetic mark that is known to be downregulated by p53 via the repression of lysine methyltransferase SUV39H1 activity[23]. This suggests that some of the peaks in cluster 4 may be p53 targets that are active in the episomal STARR-seq assay but repressed in vivo.

To extend the analysis of the properties of active enhancers to other cell types, similar clustering was performed for the STARR-seq and epigenetic data from HepG2 and RPE1 cell lines, for both of which

four clusters were determined to be optimal (Supplementary Fig. 3a). Both HepG2 and RPE1 showed relatively similar clustering patterns, with cluster 1 in both enriched for canonical active enhancer profiles (H3K4me1, H3K27ac) and cluster 2 for promoter-specific histone modifications (H3K4me3, H3K27ac) (Supplementary Fig. 3a). Clusters 3 and 4 in both cell lines were predominantly silenced, but showed some differences in the silencing mechanisms. In HepG2 cells, cluster 3 mostly showed CpG methylation but little enrichment of repressive histone marks such as H3K9me3 and H3K27me3, whereas cluster 4 was highly enriched for H3K9me3 but low in H3K27me3 or CpG methylation. In RPE1 cells, cluster 3 showed a minor enrichment of active epigenetic signals along with p53 occupancy, and cluster 4 was enriched for H3K9me3, H3K27me3, and CpG methylation (Supplementary Fig. 3a). These differences could indicate different strategies for TE silencing in different cell types, as demonstrated for example by the higher cluster-specific enrichment of H3K9me3 in HepG2 cells compared to GP5d and RPE1 cell lines.

Since GP5d STARR-seq peaks within clusters 1, 3, 4, and 5 showed a major overlap with TEs (Supplementary Fig. 2c), we next analyzed the cluster-specific enrichment of TEs at the subfamily level (Fig. 2c). In GP5d cluster 4, almost all peaks overlap a TE, which is reflected in the high relative enrichment of the subfamilies (Fig. 2c, Supplementary Fig. 2c). Peaks in clusters 4 and 5 overlap with genomic p53 binding and we found that they are specifically enriched for TE subfamilies such as MER61E and multiple LTR10 subfamilies (Fig. 2c) that have been previously reported to harbor near-perfect p53 binding motifs[24]. The strong STARR-seq signal observed from these clusters is consistent with the previous reports that have demonstrated strong activity in a STARR-seq assay from p53-binding enhancers[17,25].

HepG2 and RPE1 showed a very similar pattern of TE enrichment. Compared to GP5d, HepG2, and RPE1 showed fewer TE subfamilies enriched in cluster 1, with only five and two enriched TE subfamilies in these cell lines, respectively (Supplementary Fig. 3b). This might suggest that loss of silencing mechanisms activates TE enhancers with a higher frequency in colorectal cancer which is known to be a cancer type with high transpositional activity[12]. Similar to GP5d, cluster 1 in HepG2 and RPE1 was enriched for the JUN motif, while cluster 2 was enriched for promoter-specific KLF/SP as well as YY family zinc finger factor motifs (Supplementary Fig. 3c). Remarkably, clusters 3 and 4 in both HepG2 and RPE1 cells were enriched for p53-specific subfamilies such as MER61 and LTR10 (Supplementary Fig. 3b), which were also enriched in GP5d clusters 3–5. This indicates that common p53-specific TE subfamilies, occupied by p53, show consistent STARR-seq activity in different cell lines.

The strongest enrichment for active enhancer marks in GP5d cells was observed from peaks within cluster 1 (Fig. 2a). Interestingly, cluster 1 showed specific enrichment of all MER11 subfamilies, MER11A, MER11B, MER11C, and MER11D (Fig. 2c), indicating that they may have an active role as enhancers in GP5d cells. GP5d cluster 3, on the other hand, harbors peaks that are mostly silenced in the endogenous genomic context. The TE subfamilies that were found to be enriched within this cluster were similar to all other clusters, including MER11, MER61, and LTR10 subfamilies (Fig. 2c), suggesting that the members of the same subfamilies can show different levels of enhancer activity and be under different epigenetic contexts in the same cell. Taken together, by integrating the unbiased enhancer activity data from episomal STARR-seq assay to epigenetic data at corresponding genomic loci, we have identified specific TE subfamilies that are either repressed or active enhancers in human colon and liver cancer cells.

## p53 knockout reduces enrichment of specific TE subfamilies in GP5d cells

Due to the known p53-specificity of some TE subfamilies[24] that was corroborated by our findings of high enrichment of MER61 and LTR10

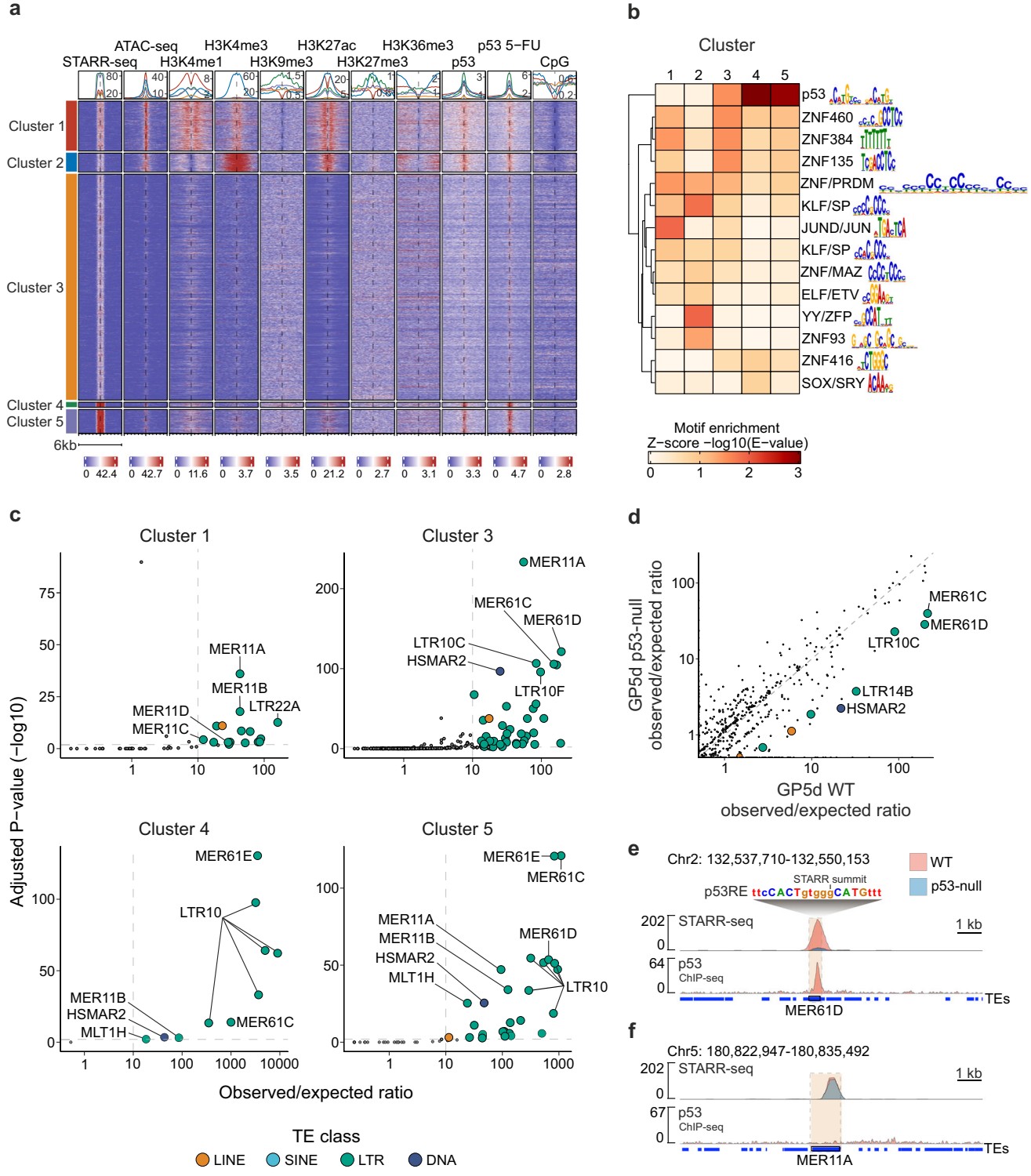

**TE class**
- 🟠 LINE
- 🔵 SINE
- 🟢 LTR
- 🔵 DNA

elements within GP5d clusters 4 and 5, we set out to specifically study the role of p53 in TE-derived enhancer activity. For this, we analyzed previously published STARR-seq data from p53-null (p53-KO) GP5d cells, resulting in 13,349 active enhancer peaks in the absence of p53[17]. We observed that depletion of p53 in GP5d cells led to a significant decrease in enrichment for nine TE subfamilies (one-sided Fisher's test FDR < 0.01; Fig. 2d, e), including MER61 and LTR10 subfamilies. LTR was also the only TE class that was significantly enriched for p53 binding in ChIP-seq data (BH-adjusted one-sided binomial test FDR = 0.0094, Supplementary Fig. 4a). Overall, the enrichment of TE subfamilies within the STARR-seq peaks correlated with the proportion of

their genomic copies harboring p53 binding motifs (Supplementary Fig. 4b), and as expected, the TE subfamilies affected by p53 depletion were highly p53-specific: for example, 53.4% of MER61C copies harbor a p53-response element (p53RE)[24]. However, not all p53-specific TE subfamilies were affected by p53 depletion. For example, MER61E with 35.4% of its genomic copies harboring a p53 binding motif showed a reduction from 80-fold in WT to 57-fold in p53-null, but the effect was not statistically significant. This could be due to the redundancy of the p53 family members, suggesting that other members can bind and control these elements upon depletion of p53. TE subfamilies significantly affected by depletion of p53, e.g. MER61C and MER61D,

**Fig. 2 | TEs are enriched for distinct epigenetic signatures and TF binding motifs in colon cancer cells. a** K-means clustering of all STARR-seq peaks in GP5d with signal for open chromatin (ATAC-seq), ChIP-seq for epigenetic marks and transcription factor binding (H3K4me1, H3K4me3, H3K9me3, H3K27ac, H3K27me3, H3K36me3, p53, and 5-fluorouracil-treated p53), and CpG methylation (NaNOMe-seq). Five clusters was optimal for clustering as determined by elbow plotting (see Supplementary Fig. 2a). **b** TF motif enrichment at all STARR-seq peaks cluster (*n* of peaks shown in Supplementary Fig. 2b) within each cluster from **a**. After performing motif enrichment analysis for individual motifs, similar motifs were combined into motif clusters according to ref. 108. The representative TF family and motif are shown for each clustered motif. **c** Enrichment of TE subfamilies within STARR-seq peaks in each cluster. Cluster 2 is omitted due to no significant enrichment. Observed/expected ratio on *X* axis is calculated as the count of STARR-seq summits overlapping TE subfamilies divided by the mean count of 1000 repetitions of randomly shuffled STARR-seq peak summits overlapping TE subfamilies. Adjusted

*P* value on *Y* axis is the −log10-transformed BH-adjusted one-sided binomial test FDR. Dashed line on *Y* axis represents the significance limit (FDR = 0.01) and on *X* axis the observed/expected ratio of 10 used as the limit of significance for STARR-seq peak summit overlap with TE subfamilies. **d**, Comparison of TE subfamily enrichment in all STARR-seq peak summits between wild-type (WT) and p53-null GP5d cells. Colored points mark differentially enriched TE subfamilies (BH-adjusted one-sided Fisher's exact test FDR < 0.01). Significant subfamilies with an observed/expected ratio >10 are labeled. **e**, Genome browser view of an example STARR-seq peak overlapping a MER61D element. The tracks for STARR-seq and p53 signals from the two cell lines are shown using the same scale and overlayed on top of each other, showing a loss of STARR-seq signal and p53 binding at the locus. **f** Genome browser view of STARR-seq peak overlapping a MER11A element, showing no p53 binding and equal STARR-seq signal in both WT and p53-null GP5d. Source data are provided as a Source Data file.

showed a more complete loss of STARR-seq signal, with a large proportion of the TE copies losing STARR-seq signal completely compared to the non-significant subfamilies, e.g. MER61E, that mostly showed a reduction of the signal (Supplementary Fig. 4c). In conclusion, our results indicate that p53 controls TEs in a highly specific manner, and it is not associated with all TE subfamilies, as shown for example for MER11A (Fig. 2f).

**STARR-seq reveals common and cell type-specific TE enhancers**
To further interrogate if TE-derived enhancers are largely similar between cell types owing to their repetitive nature, we compared the enriched TE subfamilies for enhancer activity between GP5d and HepG2 cells. In GP5d cells, we identified 59 significantly enriched subfamilies within the STARR-seq peaks overlapping TEs (minimum of five STARR-seq summit overlaps per subfamily, BH-adjusted one-sided binomial test FDR < 0.01) (Fig. 3a, Supplementary Data 4). Of these, 53 were from the LTR class, four from the LINE class, and one each from the DNA and SINE class of TEs. In HepG2 cells, the analysis of STARR-seq peaks overlapping TEs revealed a similar pattern, with 52 TE subfamilies significantly enriched (Fig. 3a, Supplementary Data 4), majority of which (42) belong to the LTR class, five to the DNA class, four to the LINE class, and one to the SINE class. Of the significantly enriched TE subfamilies, 26 were common to both GP5d and HepG2 cells (BH-adjusted one-sided binomial test FDR < 0.01 in both, BH-adjusted two-sided Fisher's test FDR < 0.01) (Fig. 3a, *left panel*; Supplementary Data 4). The common subfamilies that showed the highest enrichment in both cell lines, such as MER61C, MER61E, and LTR10B1, were highly specific for p53[24]. This was confirmed by motif analysis that showed a high enrichment of p53 motifs within the common TE subfamilies (Fig. 3b). The observed/expected ratio of LTRs correlated in both cell lines with the percentage of genomic copies in a subfamily containing p53REs (Supplementary Fig. 5a; GP5d Pearson *R* = 0.82, *p* < 2.2e-16, HepG2 Pearson *R* = 0.76, *p* = 2.2e-16). In conclusion, these results indicate that p53-bound TE enhancers are conserved in cancer cell lines with wild-type p53.

Next, we analyzed cell type-specificity of TE enhancers and observed 18 TE subfamilies in both GP5d and HepG2 that were differentially enriched (BH-adjusted one-sided binomial test FDR < 0.01 in at least one cell line, BH-adjusted two-sided Fisher's test FDR < 0.01) (Fig. 3a, *right panel*; Supplementary Data 4). On average, differentially enriched subfamilies were evolutionarily younger than the commonly enriched elements between the two cell lines (Supplementary Fig. 5b). Specifically, THE1, MER44, MER52 and LTR12 subfamilies were over-represented in HepG2 cells and LTR14, MER11 and LTR7 subfamilies in GP5d cells (Fig. 3a, c; Supplementary Data 4). As MER11 elements have been previously shown to be active enhancers in embryonic development[26], we analyzed the enrichment of the STARR-seq peak summits overlapping MER11 elements in 127 Roadmap tissues[27] and

discovered that the TE copies that are active in GP5d cells reside in open chromatin regions exclusively in embryonic stem cells (ESC), induced ESCs and induced pluripotent stem cells in Roadmap tissues (Supplementary Fig. 5c). This suggests that cancers may utilize the same TEs that have been exapted as enhancers in embryonic development, possibly contributing to the dedifferentiated cell state of cancers[28].

We also compared the enrichment of TEs between GP5d and RPE1 as well as HepG2 and RPE1 cells (Supplementary Fig. 5d). We discovered that RPE1 had a relatively low enrichment of differential TEs, with only one TE subfamily, LTR26, differentially enriched vs. GP5d and two subfamilies, LTR14B and HSMAR2, differentially enriched vs. HepG2. LTR26 was also enriched in the RPE1 cluster 1 (Supplementary Fig. 3b), suggesting that TE exaptation into cell type-specific enhancers is relatively rare in noncancerous ectodermal cells.

Interestingly, the differentially enriched TE subfamilies also showed over-representation of distinct TF motifs (Fig. 3b). For example, TFAP2 motifs were enriched within MER11 subfamilies in GP5d and NFY motifs within LTR12 subfamilies in HepG2 cells, suggesting that distinct TFs bind to TEs in a cell type-specific manner. To confirm that the TE enhancers identified from STARR-seq are bound by the TFs suggested by the motif analysis, we mapped the ChIP-seq data for TFAP2A in GP5d cells, for NFYA in HepG2 cells, and for p53 in both cell lines to the TE-overlapping STARR-seq peaks. In good agreement with the motif enrichment analysis, we observed p53 binding mostly at the common TE elements in both GP5d and HepG2 cells (Fig. 3d). TFAP2A was almost exclusively bound to GP5d-unique TEs in GP5d cells and NFYA preferentially bound to HepG2-unique TEs in HepG2 cells (Fig. 3d). Taken together, these results indicate that TEs show cell type-specific enhancer activity that is mediated by the binding of distinct TFs.

To study whether cancer type-specific TE enrichment similar to cancer cell lines can be observed in human patient tumors, we analyzed pan-cancer ATAC-seq data for tumor samples from 23 cancer types in The Cancer Genome Atlas (TCGA) datasets[29]. We discovered that LTR class TEs were predominantly enriched within open chromatin regions in most cancer types (Supplementary Fig. 6a, b). Pan-GI cancers (colorectal, esophageal, and stomach adenocarcinomas) were commonly enriched for several TE subfamilies, many of which were also enriched in GP5d colon cancer cells, such as MER11A, LTR10C, LTR10F, and LTR14C (Supplementary Fig. 6a, c). In total, 11 out of the 18 GP5d-specific subfamilies (c.f. Fig. 3a) were also enriched in TCGA colorectal adenocarcinoma data, suggesting that the subfamilies identified in GP5d are also active in human tumor samples. In general, TE enrichment clustered remarkably well together by cancer organ system of origin and by tumor histology (Supplementary Fig. 6a, c), suggesting cancer type-specific TE activation and supporting our findings from the cancer cell lines.

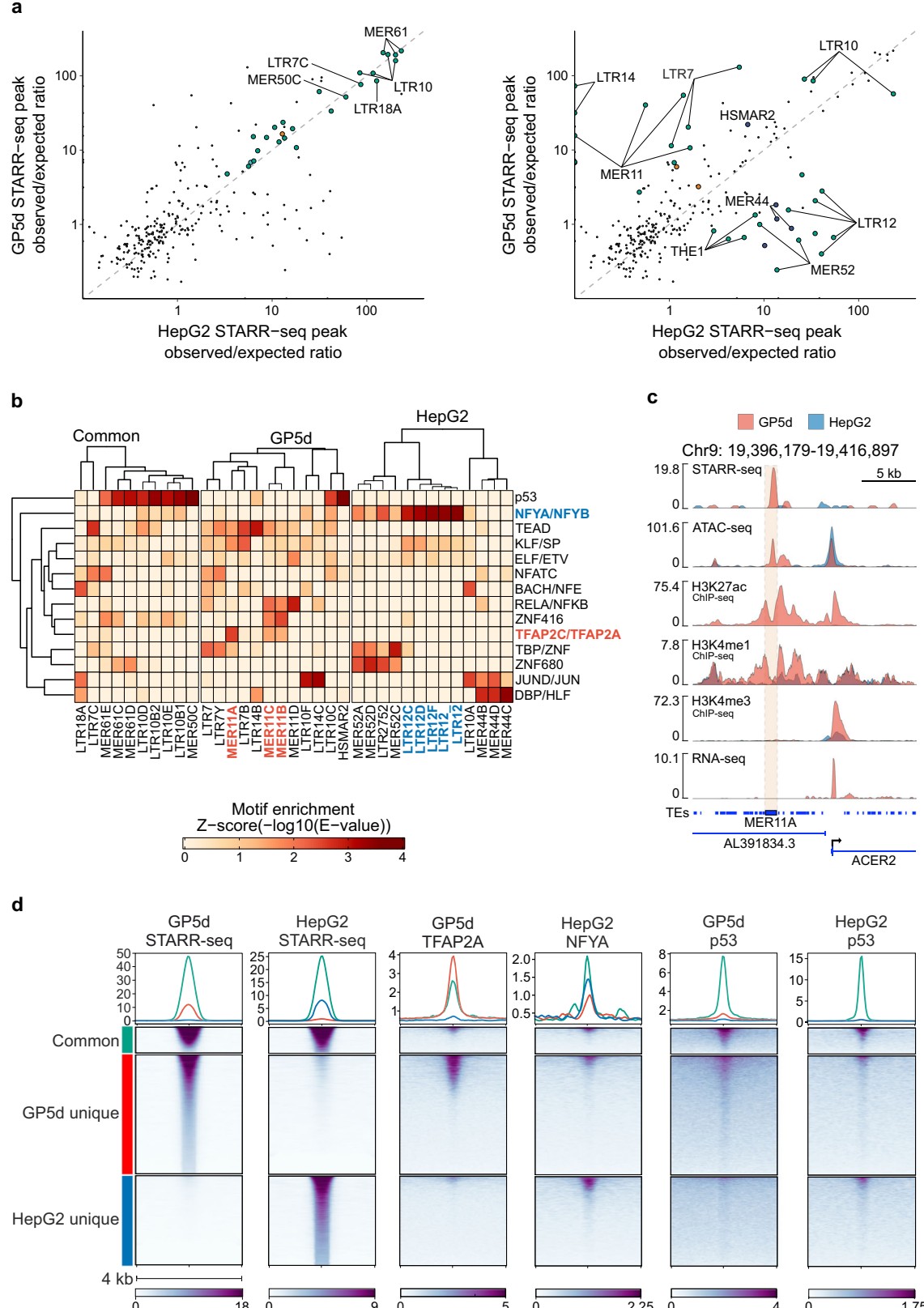

### Distinct TFs regulate cell type-specific transcriptionally active TE enhancers

To delineate TF binding dynamics at the TE subfamily level, we analyzed ChIP-seq signal at the specific LTR subfamilies that were overrepresented in distinct STARR-seq peak clusters in GP5d cells (c.f. Fig. 2a, c). As seen from the clustering, subfamilies within the MER family showed different epigenetic signals as MER11 elements were associated with classical enhancer marks and MER61 elements with p53 binding. Analysis of ChIP-seq data at these loci revealed distinct TF occupancy: TFAP2A is highly enriched at MER11B (Fig. 4a) and MER11A (Supplementary Fig. 7a) loci, whereas MER61E and MER61C are highly specific for p53 binding (Fig. 4b; Supplementary Fig. 7b).

**Fig. 3 | STARR-seq reveals common and cell type-specific differentially enriched TE subfamilies. a** Enrichment of TE subfamilies within all STARR-seq peaks in GP5d and HepG2 cells. Left panel: the subfamilies that are significantly enriched in both cell lines (BH-adjusted one-sided binomial test FDR < 0.01 in both, STARR-seq peak summit overlaps per subfamily ≥ 5) and not significantly differentially enriched between the cell lines (BH-adjusted two-sided Fisher's exact test FDR > 0.01). Right panel: the differentially enriched TE subfamilies between the cell lines (BH-adjusted two-sided Fisher's exact test FDR < 0.01, STARR-seq peak summit overlaps per subfamily ≥ 5). TE subfamilies are labeled as one group, e.g., MER11 group contains MER11A, MER11B, MER11C, and MER11D subfamilies. Supplementary Data 4 lists all the enriched subfamilies. **b** Motif enrichment for individual TE subfamilies. All significant common and differentially enriched TE subfamilies in GP5d and HepG2 that overlapped a STARR-seq peak (from Fig. 3a) were analyzed by taking the full sequences of the individual TEs with an overlapping STARR-seq peak and performing motif analysis for the sequences in each individual subfamily separately.

After motif analysis, the enriched motifs were grouped into clusters of similar binding motifs[26] (see Methods for details). The minimum E-value found for an individual TF for each motif cluster was plotted in the final figure. All individual motif hits are listed in Supplementary Data 5. **c** Genome browser snapshot of a STARR-seq peak overlapping a MER11A element, showing STARR-seq and open chromatin signals as well as ChIP-seq track for canonical histone marks of active enhancers (H3K27ac, H3K4me1) in GP5d and HepG2 cells. The active enhancer marks are observed specifically in GP5d cells whereas the HepG2 cells show negligible signals at this locus. **d** Heatmap showing signals for GP5d STARR-seq, HepG2 STARR-seq, TFAP2A and p53 ChIP-seq tracks from GP5d cells, and NFYA and p53 ChIP-seq tracks from HepG2 cells for three groups of STARR-seq peaks overlapping with TEs: shared between GP5d and HepG2 (Common, $n = 2202$), unique to GP5d ($n = 9500$), and unique to HepG2 ($n = 7147$). Source data are provided as a Source Data file.

Moreover, TF motif enrichment analysis suggested that HepG2-unique LTR12C elements are controlled by NFYA (c.f. Fig. 3b), and the analysis of NFYA ChIP-seq confirmed the binding of NFYA at these loci in HepG2 cells (Fig. 4c). These results indicate that distinct TFs can bind specific TE subfamilies and utilize them as enhancers in a tissue-specific manner.

To confirm that the TEs overlapping STARR-seq peaks are functionally active enhancers in vivo, we utilized the data for nascent transcriptional activity, TT-seq from GP5d cells[30] and GRO-seq from HepG2 cells[31]. We observed divergent transcript initiation in GP5d cells at loci where STARR-seq peaks overlap with MER11B elements bound by TFAP2A (Fig. 4d). Similarly, STARR-seq peaks in HepG2 cells overlapping LTR12C elements and bound by NFYA were transcriptionally active (Fig. 4d). Moreover, MER11B elements in GP5d and LTR12C elements in HepG2 cells show stronger nascent transcriptional activity as compared to the cluster 1 enhancers of GP5d and HepG2, respectively (Supplementary Fig. 7c, d). Next, we correlated the nascent transcriptional activity to TFAP2A binding at MER11B TE-STARR-seq peaks representing different clusters from Fig. 2a. In agreement with the STARR-seq signal and TFAP2A occupancy, TT-seq signal was strong in cluster 4, moderate in cluster 5 and weak in cluster 1 (Supplementary Fig. 7e). Similarly, TT-seq signal correlated strongly with p53 binding in three different clusters of MER61E-enriched TE-STARR-seq peaks (Supplementary Fig. 7f). These results indicate that enhancer activity measured from the episomal STARR-seq assay correlates well with in vivo TF occupancy and nascent transcriptional activity around these TE-overlapping enhancers.

As most copies of TEs in the genome are truncated, it is difficult to distinguish transcriptional activity between autonomously expressed full-length TEs and truncated TE fragments that can still function as enhancers. To avoid this bias, we filtered unique reads from ATAC-seq, ChIP-seq, and TT-seq/GRO-seq that mapped to annotated LTR elements and re-aligned the filtered reads to LTR consensus sequences (Supplementary Fig. 8a, b). We observed that accessible chromatin regions with enrichment for TFAP2A binding at MER11B and for NFYA binding at LTR12C elements associated with signals of divergent nascent transcription typical for enhancer RNAs as seen from GP5d TT-seq data and HepG2 GRO-seq data, respectively (Fig. 4e), further supporting the role of these specific subfamilies as cell type-specific enhancers derived from repetitive elements. Fig. 4f shows a representative example of transcriptionally active NFYA-bound LTR12C element located upstream of the *ZFAND2B* gene. NFYA is known to bind to promoter-proximal regions and to recruit pre-initiation complex to TSS[32], and the active transcription of the *ZFAND2B* gene in HepG2 cells is confirmed from the RNA-seq data (Fig. 4f). In conclusion, the clear position-specific enrichment of NFYA with respect to the signal of divergent transcription from GRO-seq data in HepG2 cells indicates that these repeats are functional enhancers resulting in enhancer RNA production (Fig. 4d–f).

Since the LTR12C elements showed higher enhancer activity only in HepG2 cells (c.f. Figs. 3a and 4d), we postulated that they might be epigenetically repressed and thus inactive in GP5d cells. To test this, we exposed GP5d cells to small molecule inhibitors for two epigenetic modifier enzymes, DNA methyltransferase (DNMT) and histone deacetylase (HDAC) followed by RNA-seq to specifically measure their effect on TE expression. Interestingly, DNMT and HDAC inhibition significantly upregulated LTR12C expression in GP5d cells at subfamily level (Fig. 4g, *left panel*) as well as locus level (Fig. 4g, *right panel*). To study the potential gene regulatory effect of these LTR12C-derived enhancers, we analyzed the expression of genes in the vicinity of LTR12C elements (±50 kb) from the RNA-seq data upon DNMT and HDAC inhibition. Interestingly, 89 out of 264 LTR12C-adjacent genes were differentially expressed, 70 of which were upregulated (Fig. 4h). These results indicate that epigenetic de-repression—such as hypomethylation that is a general feature of human cancers—can lead to specific activation of distinct TEs and possible transactivation of adjacent genes, demonstrating how cancer cells can utilize TEs as de novo enhancers resulting in altered transcriptome profiles in the affected cells.

## The effect of DNA methylation on TE enhancer activity

DNA methylation is an important mechanism that suppresses TE activity in the genome. To study the effect of CpG methylation on TE enhancer activity in an unbiased manner, we performed genomic STARR-seq with in vitro-methylated libraries in HepG2 cells and compared enhancer activity of TEs by using our previously published STARR-seq data from HepG2 cells[17]. In total, 6568 peaks were called for the methylated library. Interestingly, we found that DNA methylation significantly repressed specific TE subfamilies, such as L1PA3, THE1B, and THE1C (Fig. 5a–c). Motif enrichment analysis within the active THE1B and THE1C elements revealed several TF motifs harboring CG dinucleotides, such as BANP/ZBTB33 and HOXC13 (Fig. 5d). Importantly, in the motif enrichment analysis the active THE1B and THE1C elements that overlap with STARR-seq peaks were compared to elements of the same subfamilies that do not overlap with a STARR-seq peak, suggesting that regulatory differences can exist between active and silent TEs even within TE subfamilies (Fig. 5e). Some homeodomain TFs such as HOXB13 can bind to both methylated and unmethylated motifs with a higher affinity to methylated CG[33], but further studies have suggested more complex interaction between HOXB13 and methylated DNA[34]. Interestingly, in the episomal STARR-seq assay, the HOXC13 motifs with TCG residues were enriched among the enhancers whose activity decreased upon DNA methylation (Fig. 5d). Moreover, out of the two TFs that can bind the CGCG sequence enriched within the THE1 elements, ZBTB33 has been shown to bind methylated CG[35] and BANP to non-methylated CG[36], highlighting the importance of studying the effect of DNA methylation on TE activity in a locus and context-specific manner. Finally, to analyze

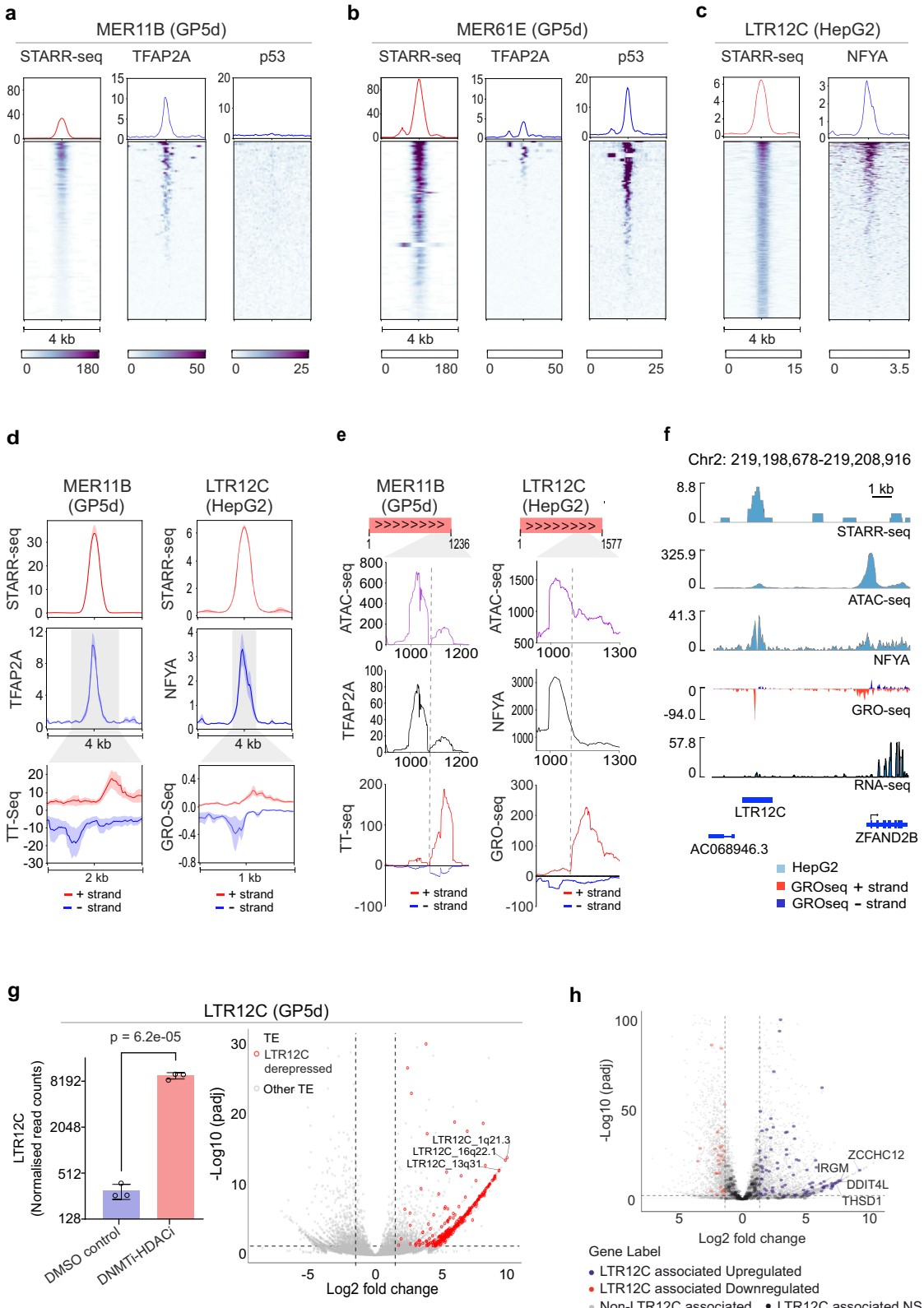

the methylation of THE1 elements at endogenous loci we utilized long-read single-molecule nanopore sequencing data, showing that genomic regions corresponding to active THE1-enhancers in HepG2 cells were less frequently methylated in HepG2 cells compared to GP5d cells (Fig. 5f). Taken together, our results suggest that overall, the effect of DNA methylation on STARR-seq activity is modest. However, few specific TE subfamilies are affected by DNA hypomethylation, and their

enhancer activity is controlled by TFs with differential binding preferences for methylated DNA.

## In silico prediction of TE enhancer-gene contacts show widespread changes in gene expression

To analyze the effect of TE enhancers on gene expression, we utilized the activity-by-contact (ABC) model[37] for in silico prediction of

**Fig. 4 | Distinct TFs regulate cell type-specific transcriptionally active TE enhancers. a, b** Heatmap of STARR-seq and ChIP-seq signals for TFAP2A and p53 at MER11B (**a**) and MER61E (**b**) elements overlapping a STARR-seq peak in GP5d cells. **c** Heatmap of STARR-seq and ChIP-seq signals for NFYA at LTR12C elements overlapping a STARR-seq peak in HepG2 cells. **d** Left panel: metaplots of STARR-seq, TFAP2A ChIP-seq, and TT-seq signals at MER11B elements overlapping a STARR-seq peak in GP5d cells. Right panel: metaplots of STARR-seq, NFYA ChIP-seq, and GRO-seq signals at LTR12C elements overlapping a STARR-seq peak in HepG2 cells. All metaplots show the average signal with standard error. **e** ATAC-seq, TFAP2A ChIP-seq, and TT-seq reads in GP5d mapped to MER11B elements overlapping STARR-seq peaks were extracted and mapped to the MER11B consensus sequence. Left panel shows metaplots of the signals in a region from 900 to 1236 bp of the MER11B consensus sequence. Similarly, ATAC-seq, NFYA ChIP-seq, and GRO-seq reads in HepG2 mapped to LTR12C elements overlapping a STARR-seq peak were extracted and mapped to the LTR12C consensus sequence. Right panel shows metaplots of the signals in a region from 900 to 1300 bp of the LTR12C consensus sequence. **f** Genome browser snapshot showing STARR-seq, ATAC-seq, NFYA ChIP-seq, GRO-seq, and RNA-seq signal at a LTR12C element overlapping a STARR-seq peak in HepG2 cells. **g** Changes in TE expression in GP5d cells upon DNMT and HDAC co-inhibition on a subfamily and locus level (see Methods for details). Left panel: the normalized RNA-seq read counts for the LTR12C subfamily with and without the inhibitor treatment (mean ± SD, individual data points for three biological replicates shown as dots; two-sided unpaired *t* test). Right panel: volcano plot of differentially expressed TEs after the co-inhibition. 498 upregulated LTR12C elements are highlighted in red. **h** Volcano plots representing gene expression changes for LTR12C-associated genes upon inhibition of DNMT and HDAC in GP5d cells. Analysis of differentially expressed genes in GP5d wild-type cells with DNMT and HDAC inhibition revealed significant upregulation of genes in the vicinity of LTR12C elements (±50 kb). Source data are provided as a Source Data file.

genomic contacts between identified TE enhancers and potential target genes (see Methods). In total, the model predicted 29,967 contacts in GP5d cells, 1206 of which overlapped with a STARR-seq peak summit and were selected for further analysis (Supplementary Data 6). Of these, 486 overlapped with a TE. As expected, most of the predicted contacts mapped to cluster 1 (c.f. Fig. 2a) with almost half of the STARR-seq peaks in cluster 1 showing at least one predicted enhancer–target gene contact. This is consistent with the active epigenetic signals (ATAC+, H3K27ac+) enriched in cluster 1 which are also used by the ABC model for contact prediction (Supplementary Fig. 9a), speaking for the activity of these TE elements in GP5d cells.

Since the MER11 elements were strongly overrepresented and active in several clusters in GP5d cells (c.f. Figs. 2c, 3a, 4a, 4d), we analyzed if the genes predicted to be connected to these elements show differential expression between GP5d and HepG2 cells. We found that the majority of the predicted MER11 target genes were overexpressed in GP5d cells in comparison to HepG2 (Fig. 6a). The expression of MER11-associated genes was also higher in GP5d colon cancer cells compared to normal colon epithelium HCoEpiC cells (Supplementary Fig. 9b), supporting the observation that MER11 subfamilies can function as active enhancers in GP5d cells. For example, a MER11B element in chromosome 3 with a strong STARR-seq signal, active enhancer marks, and TFAP2A binding in GP5d cells has a predicted association with several genes that are strongly expressed in GP5d cells (Supplementary Fig. 9d). In HepG2 cells, however, the MER11B-enhancer is not active, and the expression of these genes is low despite the active epigenetic marks at their promoters (Supplementary Fig. 9d).

As TEs enriched within many STARR-seq peak clusters are highly p53-specific, such as MER61 and LTR10 subfamilies, we studied the expression of genes with a predicted contact to these TEs. For this, we compared gene expression between wild-type GP5d and HepG2 cells and observed that the genes with predicted association to MER61 and LTR10 subfamilies also showed upregulation in GP5d cells (Fig. 6b), consistent with the higher enrichment of active enhancer features such as ATAC-seq signal and ChIP-seq signals for H3K27ac and H3K4me1 observed in GP5d cells compared to HepG2 cells at some of these elements (Supplementary Fig. 9e). Previously, it has been shown that despite the presence of strong p53REs in TEs, p53 binding rarely resulted in transactivation of genes[38]. We also show that relatively few of the highly enriched p53-specific TEs had a predicted contact to a gene, but some of them were associated with an overexpressed gene and active enhancer marks in GP5d cells (Fig. 6b, Supplementary Fig. 9e). When comparing the expression of the genes with predicted contact to a p53-associated TE between wild-type and p53-null GP5d cells, only few of the genes showed significant difference in expression (Supplementary Fig. 9c). Primary cilia formation gene, *PIFO*, had a predicted contact to a p53RE-containing LTR10D element and showed a significantly altered expression after

p53 depletion as well as in GP5d cells compared to HepG2 (Fig. 6b, Supplementary Fig. 9c, e). In GP5d, the LTR10D element had an active enhancer profile, whereas in HepG2 cells the H3K27ac signal was lacking, suggesting a poised enhancer state[39] and consequently lower expression of the gene (Supplementary Fig. 9e). This suggests that p53RE-containing TEs may also have a minor role in regulating gene expression as a response to p53.

## MER11-derived STARR-seq enhancers transactivates nearby genes in GP5d cells

To establish the *cis*-regulatory activity of MER11 elements in GP5d cells, we deleted three MER11B elements with strong STARR-seq activity by using CRISPR-Cas9 (Fig. 6c). We targeted each element using Cas9 together with specific pairs of guide RNAs for each flank of MER11B overlapping STARR-seq peaks (Supplementary Data 9). We generated clonal cell lines with homozygous deletions for two MER11B elements and a heterozygous clone for third MER11B element (Supplementary Fig. 11a–d). We used RT-qPCR to determine expression changes of potential target genes (predicted from ABC model) for each of the MER11B elements. In the case of MER11B element at the *CARD14* and *EIF4A3* locus (Fig. 6c, *left panel*), two independent clonal cell lines with homozygous TE deletion show downregulation of the predicted target genes upon deletion of the specific MER11B element (Fig. 6d, Supplementary Fig. 11b). Similarly, deletion of MER11B element on chromosome 2, (Fig. 6c, *middle panel*), downregulated *SPOPL* and *SPOPL-DT* but not the third nearby gene *HNMT* (Fig. 6d, Supplementary Fig. 11c). However, heterozygous deletion of MER11B element on chromosome 8, (Fig. 6c, *right panel*), did not affect expression of *NUDCD1* (Fig. 6d, Supplementary Fig. 11d). These CRISPR deletion experiments indicate that MER11B-derived enhancer elements contribute to the transcriptional activation of nearby genes in *cis*.

## Discussion

It is well-established that TEs are a rich source of gene regulatory elements[2,40] and have contributed to the evolution of gene regulatory networks[13]. However, the role and the extent of TEs as onco-exapted enhancers are less well understood. Here, we studied the functional enhancer activity of TEs in two endodermal origin cancers and a non-transformed ectodermal cell line by combining data from a quantitative whole-genome enhancer assay with extensive epigenetic analyses. The advantage of the episomal STARR-seq assay is that it enables measuring the regulatory potential of each genomic element in an unbiased manner, allowing the functional analysis of endogenously silent repetitive elements, whereas the epigenome analysis reveals the regulatory status of each element in the endogenous genomic context. We identified specific TE subfamilies that are enriched in colon and liver cancers with a notable specificity in active TE enhancers between these two cancer types, suggesting that the TE enhancers can be activated in a cell type-specific manner.

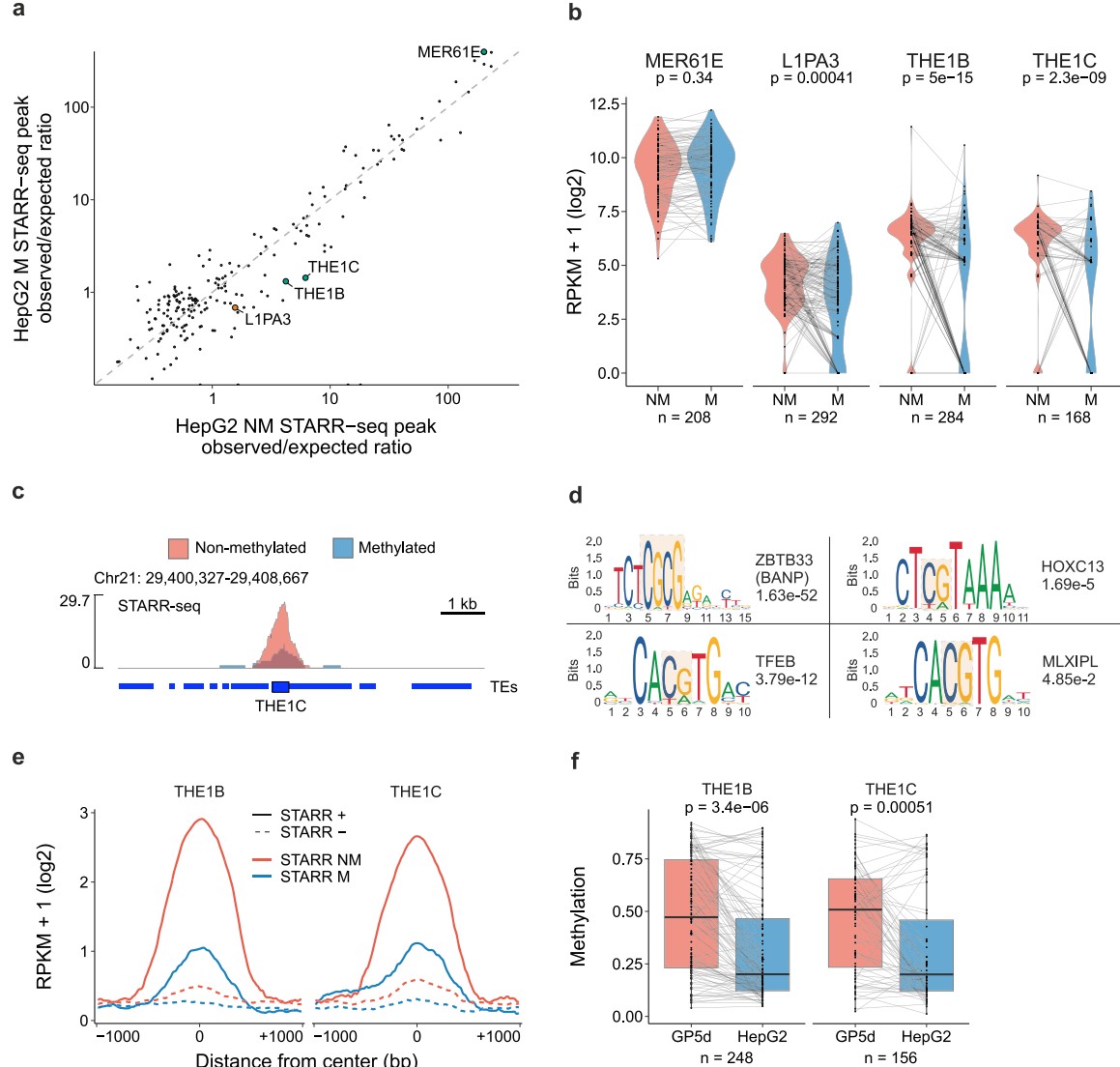

**Fig. 5 | The effect of DNA methylation on TE enhancer activity. a** The effect of methylation on enhancer activity at all STARR-seq peaks for the non-methylated (NM) and in vitro-methylated (M) genomic libraries in HepG2 cells. TE subfamilies with significant differential enrichment (BH-adjusted two-sided Fisher's exact test FDR < 0.01) are highlighted and labeled; gray points represent subfamilies that were insignificant. **b** Comparison of STARR-seq signal between NM and M libraries in HepG2 cells for the four significant TEs from **a**. STARR-seq read counts were counted at the same elements in the two libraries and RPKM-normalized. *P* values were calculated with a two-sided, paired Wilcoxon test (*n* = 208, 292, 284, and 168 for MER61E, L1PA3, THE1B, and THE1C, respectively). **c** Example genome browser snapshot of a THE1C element showing a reduction in STARR-seq signal between the NM and M libraries in HepG2 cells. **d** Motif enrichment within THE1B and THE1C elements overlapping a STARR-seq peak summit in HepG2 cells compared to non-STARR-overlapping THE1B and THE1C elements. The sequence logo, name of the TF and the E-value are from AME analysis (see Methods for details). Only significant human-specific motifs are shown (*E* value < 0.05). **e** Metaplot of RPKM-normalized STARR-seq signals using the NM and M libraries at THE1B and THE1C elements in HepG2 cells. Coverage was calculated in a ±1 kb region from the center of the elements, log2-transformed, and a rolling mean was calculated with a 25 bp window. Solid line shows the signal at the STARR-seq-overlapping THE1B and THE1C elements, and the dashed line shows the signal at 1000 randomly sampled non-overlapping THE1B and THE1C elements. **f** Boxplots of CpG methylation at THE1B and THE1C elements in GP5d and HepG2 cells analyzed from NaNOMe-seq data. Average signal is shown for all the elements overlapping a STARR-seq peak summit in HepG2. The boxplots indicate the median (center line), the third and first quartiles (box limits) and 1.5 × IQR above and below the box (whiskers). P-values are from a two-sided paired Wilcoxon test (*n* = 248 and 156 for GP5d and HepG2, respectively). Source data are provided as a Source Data file.

TEs are under strict epigenetic control in normal somatic cells through DNA methylation, repressive histone modifications, RNA-mediated silencing, and silencing mediated through TFs such as p53 and KRAB-ZNFs[21,22,41]. Two opposing hypotheses have been proposed for TE activity and exaptation based on the evolutionary age of the elements and their epigenetic status:[42] exaptation hypothesis predicts that more conserved and older TEs are more enriched for active histone marks due to their exaptation to regulatory roles during genome evolution, and defense hypothesis predicts that young TEs will be more enriched for repressive marks due to their potential disruptive activity. This classification was based on histone modifications only, whereas here we have also used nascent transcription data and unbiased enhancer activity measurements for detailed characterization of TE activity. In favor of the defense hypothesis, we observed strong STARR-seq activity, more active histone modifications, and nascent transcription at relatively young LTR subfamilies as opposed to older TE subfamilies, and the predominant adaptation of young TEs to novel regulatory elements in mammals has also been shown in a recent study[43]. However, our results highlight the context-specificity of TE activation, since different copies of the same subfamily can be associated with either active or repressed chromatin state as revealed by our detailed clustering analysis. Importantly, during tumorigenesis,

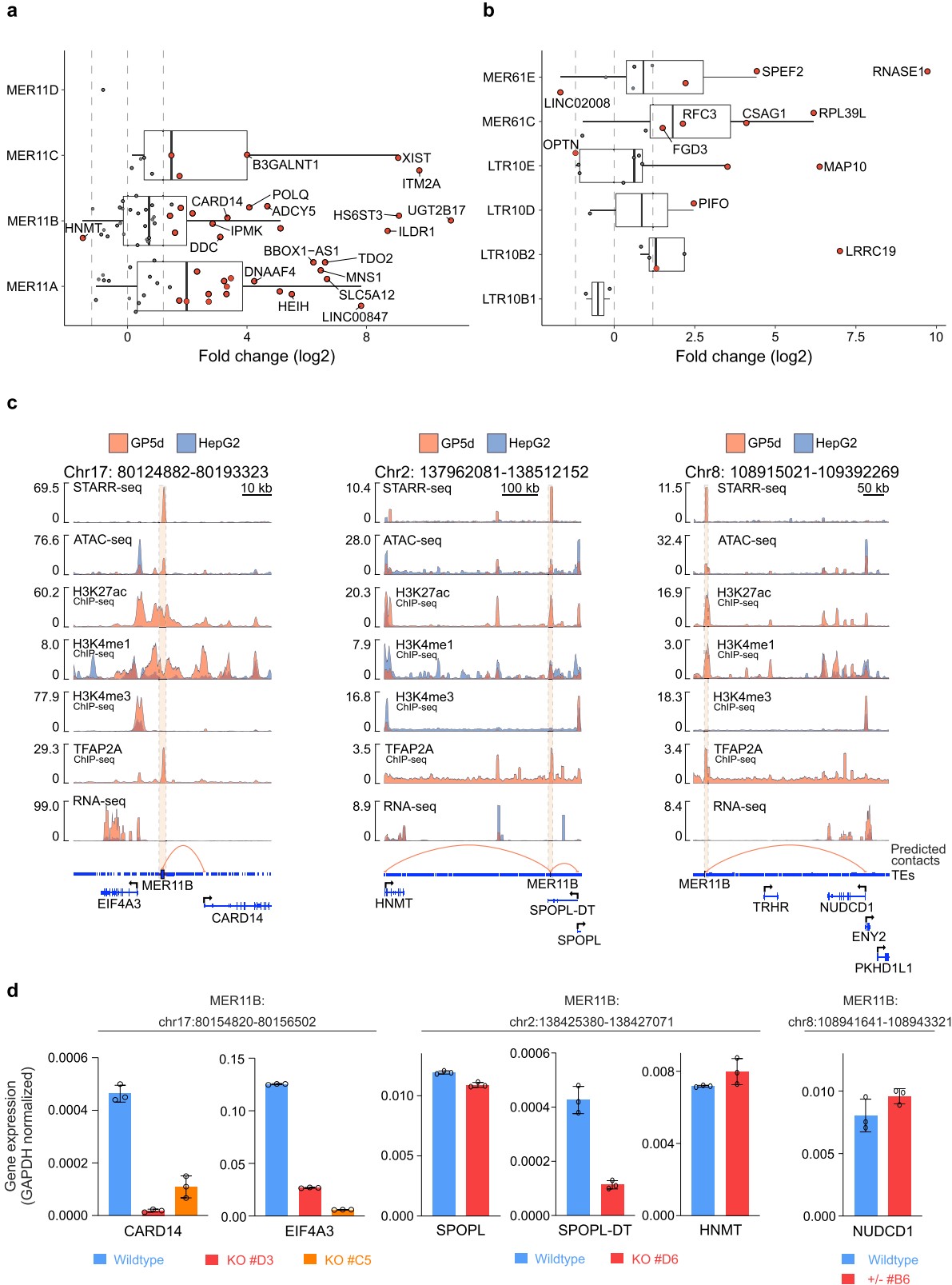

TEs can escape their silencing for example in response to epigenetic dysregulation, providing scope for onco-exaptation of previously suppressed TEs. This emphasizes the need for a detailed understanding of their suppressing and activating mechanisms in different cancer types, as done here for colon and liver cancers.

Majority of the enriched TE enhancers in both colon and liver cancer cell lines represent the ERV1 family of LTRs with significantly

more STARR-seq peaks overlapping with LTR elements than expected from a random distribution. This is consistent with earlier reports demonstrating that LTR elements contribute to ~39% of TF binding sites in both human and mouse genomes despite comprising only 8% of the human genome[40]. Transcripts originating from ERV1 elements have been detected in several cancer types, and a high level of ERV1 expression was associated with poor outcomes in kidney cancer[44,45].

**Fig. 6 | In silico-predicted TE enhancer-gene contacts show widespread changes in gene expression validated by in vivo CRISPR genome editing. a** Differential expression of genes with predicted contacts to MER11 subfamily TEs. Log2 fold change (Log2FC) is shown from RNA-seq data for GP5d vs. HepG2 cells. Red points mark significantly differentially expressed (|Log2FC|> 1.2 and FDR < 0.05), and blue points are insignificant. Log2FCs of ±1.2 and 0 are marked with dashed lines. Box-plots indicate the median (center line), the third and first quartiles (box limits), and 1.5 × IQR above and below the box (whiskers). (n = 80 predicted contacts). **b** Differential expression of genes with predicted contacts to p53-specific TEs from RNA-seq data between GP5d WT vs. HepG2 cells. Plot elements are like in **a**. (n = 32 predicted contacts). **c** Genome browser snapshots of three active MER11B elements with high STARR-seq signal in GP5d cells. Each panel shows the highlighted TE locus with signal tracks in GP5d and HepG2 cells for STARR-seq, ATAC-seq, ChIP-seq for TFAP2A and histone marks, RNA-seq, and predicted enhancer-gene contacts from the ABC analysis. *CARD14* in the left panel, *HNMT*, *SPOPL*, and *SPOPL-DT* in the middle panel, and *NUDCD1* in the right panel were predicted as gene targets for the MER11B elements. These MER11B elements mostly show STARR-seq activity and canonical epigenetic marks of enhancers in GP5d, likely reflected in the low expression of the target genes in HepG2. **d** RT-qPCR data showing changes in mRNA expression for genes with predicted contacts to active MER11B elements upon CRISPR-Cas9 mediated deletion of the elements in GP5d cells (GAPDH normalized). Two independent clones with homozygous enhancer deletion were analyzed for MER11B element at chromosome 17 in the vicinity of *CARD14* and *EIF4A3* (shown in Fig. 6c) and one for MER11B at chromosome 2 flanked by *SPOPL*, *SPOPL-DT* and *HNMT* (shown in Fig. 6c). One heterozygous deletion clone for MER11B at chromosome 8 predicted to regulate *NUDCD1* was also tested (shown in Fig. 6c). The figures show mean ± SD values for three technical replicates. Source data are provided as a Source Data file.

The widespread ERV expression detected in human cancers speaks for the global de-repression of epigenetic silencing of these elements. Interestingly, nascent transcription from enhancers has been shown to be more predictive of enhancer activity than enrichment of active histone marks[46]. Here, we showed that the enhancer activity of TEs measured using the episomal STARR-seq assay correlates well with nascent transcription and active histone marks from the respective genomic loci. Our data suggests that the same transcriptionally active TEs that evade epigenetic suppression can also be functionally active enhancers, warranting further studies about their significance in tumorigenic processes.

The active enhancers detected from GP5d colon cancer cells and HepG2 liver cancer cells using STARR-seq were enriched for distinct as well as common TE subfamilies. The common elements, such as primate-specific MER61 and LTR10 subfamilies, were strongly enriched for p53 motif. p53 is functionally active in both cell lines, and the common active TEs overlapped with p53 binding in the genome. The common elements were among the most highly enriched subfamilies in both cell lines and had the highest STARR-seq activity but only weak enrichment for the active histone marks, consistent with earlier reports from human and mouse cells[17,47]. The enrichment of TE subfamilies correlated with the proportion of their genomic copies harboring p53 binding motifs. Previous studies have shown that active and functional p53 enhancers are characterized by a single canonical p53 motif without binding of coregulatory TFs[48], commensurate with our findings in clusters 4 and 5 with strong enrichment of p53 motifs. A recent study found that p53 enhancers are also likely to be independent of interaction via Mediator[49], further supporting independent function of p53 as a transcriptional regulator.

In the context of TE regulation, both silencing and activating roles have been suggested for p53[24,50]. p53 is known to be a pioneer factor[51,52] that can preferentially bind to regions of high nucleosome occupancy[53] and closed chromatin enhancers[17]. It has been shown previously that TE-associated p53-occupied p53REs were stronger than non-TE-associated p53REs, especially at LTR10B and MER61 elements, but p53 binding at TEs did not cause changes in expression of target genes[38]. Our findings were in line with this, showing that the expression of in silico-predicted target genes of MER61 and LTR10 enhancers was largely unaffected by p53 depletion. In our study, strong p53REs were found within enhancers and repressed chromatin, possibly explaining the strong STARR signal, high H3K9me3, and low chromatin accessibility in cluster 4. In conclusion, our results support both silencing and activating functions for p53, showing that majority of the p53-enriched TEs that show enhancer activity in the episomal STARR-seq assay are associated with repressive histone marks at the endogenous chromatin loci consistent with a previous report[54], but that specific upregulation of TE-associated predicted p53 target gene, such as *PIFO*, was also detected.

Our analysis of the STARR-seq peaks also revealed active TE enhancers specific to each cancer type. MER11 subfamilies of the LTR class (MER11A, B, C, and D) were specifically active in GP5d colon cancer cells showing enrichment for TFAP2A motif and overlap with TFAP2A binding in the genome. Nascent transcriptional activity from the MER11B elements correlated with TFAP2A ChIP-seq and STARR-seq signals, further speaking for the enhancer activity of MER11 elements. Interestingly, CRISPR screen data from Cancer Dependency Map (DepMap) collection shows that TFAP2A depletion has the strongest growth inhibitory effect in GP5d cells across 55 colorectal cell lines (see Supplementary Fig. 8c) (CRISPR Chronos score = −0.2586)[55], suggesting that TFAP2A-controlled transcriptional programs are critical for GP5d proliferation. Further, our CRISPR-Cas9 deletion experiments functionally validated multiple MER11 elements with STARR-seq activity that regulates nearby genes in *cis* in GP5d cells.

In HepG2 liver cancer cells, we found that the LTR12 subfamilies were specifically enriched within the STARR-seq peaks. Consistent with a previous report, LTR12 subfamilies were highly enriched for NFY family motifs[56–59] and our data shows nascent transcriptional activity from NFYA-bound LTR12-enhancers. NFYA is essential for maintaining nucleosome-depleted regions at promoters of NFYA target genes and preventing sub-optimal transcription initiation from alternative TSSs by directing transcriptional machinery to the correct TSS[32]. The role of LTR12C as an alternative promoter[59,60] and their transcriptional repression by KRAB zinc fingers has been studied previously in early embryonic development[61]. LTR12C and other hominoid-specific LTR elements undergo H3K9me3-dependent heterochromatin remodeling during early embryonic development[62]. LTR12 retrotransposons which are the solitary LTRs of ERV9 endogenous retroviruses have also been described as enhancers in oocytes and erythroblasts[63,64]. Interestingly, we show that NFYA is associated with enhancer activity from the STARR-seq-identified LTR12 elements with high enrichment of the LTR12C within the STARR-seq peaks, raising the question of whether LTR12C elements can also function as promoter-enhancers that were reported earlier[65].

The general mechanisms for TE silencing in somatic cells include DNA methylation, histone modifications, and RNA-mediated silencing. Interestingly, our results suggest that tissue-specific TE activation is associated with specific de-repression of epigenetic silencing. For example, LTR12C elements were found to be more active in HepG2 compared to GP5d cells, but inhibition of epigenetic regulatory enzymes DNMT and HDAC resulted in strong and specific upregulation of LTR12C elements in GP5d cells. Importantly, this also led to increased expression of endogenous genes in the vicinity of LTR12C elements, suggesting that cancer cells can utilize TEs for controlling the expression of endogenous genes. Furthermore, differential CpG methylation was observed between HepG2 and GP5d cells at THE1 elements that were shown to be more active in non-methylated conditions compared to methylated DNA in HepG2 cells. Taken together, our results demonstrate how re-activation of specific TE elements can occur through destabilization of epigenetic repressive mechanisms.

Due to the inherent difficulties of mapping the short-read sequences to the repetitive sequences within TEs, the results in this

study may underestimate the actual extent of the functional repertoire of TEs. Here, we only retained uniquely mapping reads, but methods to improve read assignment to TEs[66,67] and peak calling for STARR-seq have been developed[68]. However, despite the lower mappability of evolutionarily young subfamilies, we found that relatively young LTR subfamilies were highly enriched in the STARR-seq data. Improvements in both short- and long-read sequencing technologies will benefit the future studies of TEs, reducing the limitations inherent to short-read sequencing methods[66,67] and peak calling for STARR-seq have been developed[68].

In conclusion, we studied the extent of TE enrichment in a genome-wide, unbiased functional enhancer assay together with orthogonal functional genomics methods for chromatin accessibility, histone modifications, and transcription factor binding. We found that specific TE subfamilies are highly overrepresented among the active enhancers, showing remarkable tissue-specificity that is controlled by distinct TFs. The contribution of these TE-derived enhancers to tumorigenic processes warrants further studies, but our results provide evidence of widespread exaptation of TEs for cancer-type-specific functional enhancer activity.

## Methods

### Data acquisition
All of the sequencing data and download links for annotation files used in this study are listed in Supplementary Data 8, including the relevant references and GEO/ENCODE accessions[69–71].

NCBI genome annotation files for GRCh38 were downloaded from Illumina iGenomes.

A gene annotation GTF file was acquired from Gencode Release 36 for the reference chromosomes. The GTF file was transferred into a BED file with gtfToBed.sh and TSS and gene body BED files were created with a script adapted from ref. 72.

A repeatMasker.txt (2021-09-03) file was downloaded from the UCSC table browser. Only transposable element-derived repeat classes (LINE, SINE, LTR, and DNA) were retained and a file in BED format was created from the table, totaling 4745258 annotated repeats[73]. MER11B and LTR12C consensus sequences were acquired from RepBase[74].

GRCh38 chromosome sizes file (2020-03-13) file was downloaded from UCSC.

Unified GRCh38 blacklist BED file ("ENCFF356LFX", release 2020-05-05) was downloaded from the ENCODE project.

A genome index was created with bowtie2-build, with chr1-22, X, Y, and M fasta files. Alternative, unlocalized, and unplaced alternative loci scaffolds were discarded in indexing.

Transcription factor motifs were acquired from JASPAR 2022 CORE non-redundant vertebrate annotations[75]. The position weight matrices in MEME format were used for downstream motif analyses.

Predicted LTR p53 binding site percentages were downloaded from ref. 24. TE age estimation data were downloaded from the TE-analysis pipeline[76].

### Cell culture
GP5d cells (Sigma, 95090715) were cultured in DMEM (Gibco, 11960-085) supplemented with 10% FBS, 2 mM L-glutamine (Gibco, 25030024) and 1% penicillin-streptomycin (Gibco, 15140122). HepG2 (ATCC, HB-8065) was cultured in MEM (Gibco, 11544456) supplemented with 10% FBS, 2 mM L-glutamine, and 1% penicillin-streptomycin. HCoEpiC (ScienCell, 2950) was cultured in Colonic Epithelial Cell Medium (ScienCell, 2951) as per vendor's guidelines. RPE1 cells (ATCC, CRL-4000) were cultured in DMEM F12 (Gibco, 11330-032) supplemented with 10% FBS and 1% penicillin-streptomycin (Gibco, 15140122).

All cell lines were directly obtained from trusted vendors (Sigma, ATCC, and SciencCell) and low-passage cells were used in experiments. Vendors such as ATCC perform authentication and quality-control tests on all distribution lots, so the cell lines were not re-authenticated by the user. All cell lines tested negative for mycoplasma contamination upon purchase and were routinely checked as per standard good laboratory practice.

### Genomic STARR-seq
Genomic STARR-seq experiment was performed in HepG2 and RPE1 cells as previously described[17] by using methylated and non-methylated input DNA library from ref. 17, respectively (two technical replicates, 170 million cells per replicate). One μg of input library DNA was transfected per million cells. In brief, 10 million cells were seeded per 15-cm plate in media without antibiotic a day before transfection. Plasmid DNA was mixed with transfection reagent (Transfectin (Bio-Rad, 170-3351) was used for HepG2 at a 1:3 ratio and TransfeX™ (ATCC, ACS-4005) was used for RPE at a 1:2 ratio) in Opti-MEM medium (Gibco, 11524456), incubated for 15 minutes at room temperature and added dropwise to the cells. Cells were harvested after 24 hours after transfection and total RNA isolated using the RNeasy Maxi kit (Qiagen, 75162) with on-column DNase I digestion. Dynabeads mRNA DIRECT Purification kit (Invitrogen, 61012) was used to purify poly(A) + RNA. poly(A) + RNA was treated with DNase by using TurboDNase (Ambion, AM2238) followed by purification by using RNeasy Minelute kit (Qiagen, 74204). STARR-seq reporter library was prepared by following protocol as described in ref. 17. and paired-end sequenced. No spike-in strategies for the RNA-seq data were used in this study.

### ChIP-seq, ATAC-seq, NaNoMe-seq and Hi-ChIP
ChIP-seq was performed as previously described[17] by using the following antibodies for: H3K4me3 (Sigma-Aldrich, 07-473), H3K36me3 (Diagenode, C1541092-10), TFAP2A (Abcam, AB52222), mouse IgG (Santa Cruz Biotechnology, sc-2027) and rabbit IgG (Santa Cruz Biotechnology, sc-2025). ChIP-seq was performed by using 2 μg of antibody per reaction. In brief, GP5d cells were formaldehyde cross-linked for 10 minutes at room temperature. Sonicated chromatin was centrifuged, and the supernatant was used to immunoprecipitate DNA using Dynal-bead coupled antibodies. Immunoprecipitated DNA was purified and used for ChIP-seq library for Illumina sequencing. The libraries were single-read sequenced on NovaSeq6000. H3K36me2 ChIP-seq replicate 2 was deeper sequenced on MiSeq.

ATAC-seq library was prepared by using 50000 GP5d cells by using protocol described earlier[77]. GP5d cells were washed with ice-cold PBS and cells were resuspended in 50 μl lysis buffer. Cells were incubated for 10 minutes on ice. The pellet was resuspended in 2× tagmentation buffer (Illumina kit) and incubated at 37 °C for 30 minutes. DNA was purified by using MinElute purification kit and eluted in nuclease free water. Optimal number of amplification cycles was determined by qPCR. Samples were amplified by using Nextera library preparation kit (Illumina) and sequenced paired-end.

NaNoMe-seq was performed to profile CpG methylation and chromatin accessibility (GpC methylation) in GP5d cells as described earlier[70]. GP5d cell nuclei were isolated and treated with GC methylase M.CviPI (New England Biolabs, M0227) as described[70]. Following GC methylation, DNA was isolated from nuclei by using phenol-chloroform extraction protocol, and sequencing library was prepared using the 1D genomic DNA by ligation kit (SQKLSK109) according to manufacturer's protocol and 50 fmol of adaptor-ligated genomic DNA was loaded to the flow cell for sequencing.

H3K27ac Hi-ChIP was performed as previously described in ref. 78. In brief, GP5d cells were formaldehyde cross-linked for 10 minutes at room temperature, and nuclei were isolated from cross-linked cells by 30 minutes of lysis at 4 °C. Isolated nuclei from 15 million cells were permeabilized in 0.5% SDS for 10 minutes at 62 °C and SDS was quenched by using Triton X-100 for 15 minutes at 37 °C. Chromatin was digested with MboI restriction enzyme (New England Biolabs, R0147) at 37 °C for 2 hours and then heat-inactivated at 62 °C for 20 minutes.

Chromatin was incubated with a DNA ligase (New England Biolabs, M0202) for 4 hours at room temperature for proximity ligated contact formation followed by centrifugation. Proximity ligated chromatin pellet was sonicated and immunoprecipitated using 5 µg of H3K27ac antibodies (Diagenode, C15410196). Immunoprecipitated fragments were Biotin-labeled (Invitrogen, 19524016) and captured by Dynabeads MyOne Streptavidin C1 beads (Invitrogen, 65001). Immunoprecipitated DNA was adaptor-labeled by using Tn5 transposase (Illumina) and PCR amplified. The libraries were paired-end sequenced on NovaSeq 6000.

## DNMT and HDAC inhibitor treatment and RNA-seq

DNMT and HDAC inhibition in GP5d cells were performed as described earlier[56]. GP5d cells were seeded in 6 well plates and treated with 500 nM/L 5-aza-2'-deoxycytidine (MedChemExpress, HY-A0004) or DMSO (Fisher, BP231). 5-aza-2'-deoxycytidine containing media were replaced each day for three days. SB939 was added at a concentration of 500 nmol/L (MedChemExpress, HY-13322) and cells were harvested after 18 hours. Total RNA was extracted using RNeasy Mini kit (Qiagen) according to the manufacturer's instructions in three biological replicates. RNA-seq libraries were prepared using 1 µg of total RNA input using KAPA stranded mRNA-seq kit for Illumina (Roche) as per manufacturer's instruction and paired-end sequenced on NovaSeq 6000 (Illumina).

## CRISPR-mediated knockout of MER11 elements

For CRISPR-mediated knockout of MER11 elements, two gRNAs (one specific to each flank of the MER11 element) were designed using CRISPOR v.5.01[79] and synthesized as crRNAs by Integrated DNA Technologies (IDT) (Supplementary Data 9). Early passage GP5d cells were transfected with ribonucleoprotein (RNP) complex. Equimolar ratios of target-specific crRNAs and ATTO550-tracrRNA (IDT, 1075928) were annealed. RNP complex were constituted from Alt-R S.p. HiFi Cas9 Nuclease V3 (IDT, 1081060; 1000 ng per 200,000 cells) and target-specific sgRNA (250 ng per 200,000 cells) and transfected to cells by using CRISPRMAX (Life Technologies, CMAX000003) according to manufacturer's protocol. For generating clonal cell lines from transfected cells, GP5d cells transfected with RNP complex targeting MER11 elements along with a non-transfected control were trypsinized 24 hours after transfection, washed once, and resuspended in cold PBS. ATTO550-positive transfected cells were separated from non-transfected and dead cells using flow cytometry analysis at the HiLife Flow Cytometry Unit, University of Helsinki, Finland, using BD Influx System (USB) and BD FACS software (version 1.2.0.142). Gate for sorting was set so that all non-transfected GP5d cells remained negative for ATTO550 (at 561–581 nm) and the same gate was used for sorting transfected GP5d cells so that cells positive for ATTO550 were seeded to 96 well plates (one cell per well) (Supplementary Fig. 10). After two-three-week culture, the established single cell clones were screened for homozygous deletion by rapid DNA lysis (Lucigen, QE0905T) and PCR using flanking primer pairs at the expected deletion site (Supplementary Data 9).

For analyzing the expression of potential target genes using qRT-PCR, RNA was isolated from clones with homozygous or heterozygous deletion using RNeasy Mini kit (Qiagen). cDNA synthesis was performed using the PrimeScript™ RT Master Mix (Takara, RR036A) and real-time PCR was performed using SYBR Green I Master (Roche, 04707516001) in triplicates. The primers used for each transcript are listed in Supplementary Data 9. The transcript levels of the target genes were normalized to GAPDH mRNA levels.

## Genomic STARR-seq analysis

FastQC v.0.11.9 was used for quality control and determination of read lengths of raw data[80]. For GP5d, the input plasmid control reads were trimmed with Trimmomatic v.0.39 from -76 bp to 36 bp, matching the read length of reporter cDNA to avoid biases in mappability[81]. For

HepG2, the two technical replicates were combined before alignment. Bowtie2 v.2.4.1 was used to map the paired-end STARR-seq reads uniquely to the reference human genome (hg38/GRCh38) with the (bowtie2 --maxins 1000)[82]. Duplicates were marked with Picard v.2.23.4 (MarkDuplicates -REMOVE_DUPLICATES false -ASSUME_SORTED true) and quality metrics were determined with picard CollectMultiple-Metrics (http://broadinstitute.github.io/picard/). Samtools v.1.7 was used to filter non-concordant reads and reads with a MAPQ smaller than 20 (samtools view -h -F 1024 -q 20)[83]. MACS2 v2.2.7.1 was used for calling peaks with options −f BAMPE −g hs and the STARR input as a control[84]. Peaks overlapping with ENCODE blacklisted regions were removed from the narrowPeak and summit files with bedtools v.2.29.2 (bedtools subtract -A)[85]. Genome read coverage was calculated with bedtools v.2.29.2 (bedtools coverage −pc -bg) and bedGraphToBigWig v.377 was used to create a bigwig file[86]. A RPKM-normalized bigwig file was created with deepTools v.3.5.0 (bamCoverage−binSize 50 --normalizeUsing RPKM−effectiveGenomeSize 2913022398)[87]. Final mapped read statistics are listed in Supplementary Data 7. Pearson correlation analysis between the technical replicates for RPE1 non-methylated and HepG2 methylated STARR-seq is shown in Supplementary Fig. 12a.

## ATAC-seq analysis

The raw reads were mapped with bowtie2 v.2.4.1 (bowtie2 --very-sensitive) to the reference human genome (hg38/GRCh38). Reads mapped to the mitochondrial genome were removed with removeChrom.py script (https://github.com/jsh58/harvard/blob/master/removeChrom.py). Picard v.2.23.4 was used to remove duplicates (MarkDuplicates -REMOVE_DUPLICATES false -ASSUME_SORT_ORDER coordinate) and analyze insert sizes with CollectInsertSizeMetrics. Samtools v.1.7 was used to filter reads with MAPQ smaller than 10 and remove marked duplicates (samtools view -F 1024 -b -q 10). Peaks were called with MACS2 v.2.2.7.1 (macs2 callpeak -f BAMPE -g hs−keep-dup all). Removal of blacklisted regions, coverage calculation, conversion to bigwig and normalized coverage file creation was performed as in STARR-seq data processing. Pearson correlation between GP5d ATAC-seq biological replicates is shown in Supplementary Fig. 12a.

## ChIP-seq analysis

ChIP-seq data processing was performed as in ATAC-seq except for removing the reads mapping to the mitochondrial genome and read filtering, where reads with a MAPQ smaller than 20 were removed. Pearson correlation between TFAP2A and H3K36me3 ChIP-seq biological replicates is shown in Supplementary Fig. 12a. Representative IGV screenshots for ChIP-seq and ATAC-seq replicates are shown in Supplementary Fig. 12b, c.

## Nanopore-seq analysis

The nanopore data was processed with ONT_hg_pipe v.0.1.0 (Palin 2018, unpublished, https://github.com/kpalin/). The GP5d nanopore data was basecalled with Guppy v.5.0.17 with the super-accurate basecalling model and a minimum q-score of 10. The reference genome was indexed and the basecalled reads were aligned to the reference genome with minimap2 v.2.16 (minimap2 −x map-ont)[88]. Quality controls were performed with nanoplot v.1.20.0[89] and Samtools v.1.9[71]. After alignment, methylation was called with nanopolish v.0.11.1[90]. The cpggpc_new_train branch in GitHub (https://github.com/jts/nanopolish/tree/cpggpc_new_train) was used to call both CpG and GpC methylation (nanopolish call-methylation -q cpggpc). The resulting table was processed to a BED format (mtsv2bedGraph.py -q cpggpc−nome) and to methylation frequency table formats for CpG and GpC methylation (parseMethylbed.py frequency -v -m cpg and parseMethylbed.py frequency -v -m gpc), using previously published scripts[72]. The resulting methylation tables were converted to bedGraph and bigwig formats with a custom script (mfreq_to_bw.R), utilizing bedGraphToBigWig v.377[86]. The CpG methylation frequency tables were loaded into R and smoothed with

bsseq v.1.28.0 (BSmooth ns = 50, $h$ = 1000, maxGap = 100000)[91]. The GpC methylation calling was performed for another project and was not used in this study.

## RNA-seq analysis
FastQC v.0.11.9 was used for quality control. Pseudoalignment and counting was performed with Salmon v.1.8.0[92]. Reads were also aligned with STAR v.2.5.3a[93] by using the SQuIRE pipeline v.0.9.9.92[94]. Read counts were calculated with featureCounts v.2.0.1[95]. Differential expression analysis was performed with DESeq2 v.1.32.0[96].

## Hi-ChIP-seq analysis
Hi-ChIP data for GP5d H3K27ac was processed with HiC-Pro v.3.1.0[97] with default parameters and hg38 MboI restriction sites.bed file as input. allValidPairs output from HiC-Pro was converted to a.hic file with the hicpro2juicebox.sh script from HiC-Pro, with juicer tools v.1.22[98]. KR-normalized matrices were extracted from the.hic file with juicebox_dump.py from ABC v.0.2.2[37] and powerlaw fit was calculated with the compute_powerlaw_fit_from_hic.py script.

## GP5d TT-seq and HepG2 GRO-seq
GP5d TT-seq data ("GSM4610669") was acquired under GEO accession "GSE152291"[30]. HepG2 GRO-seq raw data were downloaded under GEO accession "GSM2428726" ("SRR5109940")[31]. GRO-seq reads were trimmed to remove A-stretches originating from the library preparation by using the Trim Galore v.0.6.7. Sequence reads shorter than 25 bp and quality score <10 were discarded. GRO-seq reads were aligned to the hg38 genome assembly using bowtie2 v.2.2.5 and strand-specific bigwig files were created with Samtools v.1.9. Both intergenic as well as intragenic STARR-seq peaks were used to plot strand-specific TT-seq or GRO-seq signals.

## TEtranscripts and Telescope RNA-seq analysis
FastQC v.0.11.9 was used for quality control. Reads were also aligned with STAR v.2.5.3a[93] by using the SQuIRE pipeline v.0.9.9.92[94]. TEtranscripts v.2.2.1 was run on the SQuIRE alignment output with the following flags: –mode multi-stranded reverse. The log2 fold change values output by DESeq2 v.1.32.0 were used for TE subfamily expression analysis. Telescope v.1.0.3.1 analysis was performed on SQuIRE alignment output. The "telescope assign" command was used to quantify TE expression. The log2 fold change values output by DESeq2 v.1.32.0 were used in subsequent analysis. Nearby genes were associated with TEs with GREAT v.4.0.4[99].

## TE enrichment analysis
STARR-seq peak summits for GP5d and HepG2 were overlapped with repeatMasker annotations with GenomicRanges R package v.1.44.0[100] and the count of overlaps for each subfamily of TEs was summed. Only TE subfamilies with five or more overlaps in total were retained for further analysis.

The expected frequency of overlaps was calculated by shuffling the STARR-seq peak summits 1000 times with R package bedtoolsr v.2.29.0-3[101], excluding masked regions in from the BSgenome.Hsapiens.UCSC.hg38.masked v.1.3.993 package and keeping the shuffled features in the same chromosomes (-chrom). The mean frequency of shuffled overlaps for each TE subfamily was calculated, giving the expected frequency of overlaps. The observed/expected ratio was calculated by dividing the overlap of STARR-seq peaks by the expected overlaps for each subfamily.

Statistical significance for each TE subfamily enrichment was calculated with a one-sided binomial test with binom_test from the rstatix package v.0.7.0[102]. The number of overlaps of STARR-seq peaks was set as the number of successes, the count of all STARR-seq peaks as the number of trials, and the fraction of shuffled overlaps for a subfamily from all the shuffled overlaps was the probability of success.

P-values were adjusted with Benjamini-Hochberg correction[103]. Significance was determined as FDR < 0.01.

The differential enrichment for TE subfamilies was calculated with a two-sided Fisher's exact test with the count of overlaps for a TE subfamily as observed and the sum of 1000 shuffled overlaps as the expected frequency. P-values were adjusted with Benjamini-Hochberg correction and significance was determined as FDR < 0.01.

STARR enrichment by TE class and lineage was calculated by grouping the overlaps by class or lineage of the TE that the peak summit overlapped and calculating the enrichment against the randomly shuffled peak summits as described above. TE lineage of origin data was obtained from ref. 76.

## TCGA ATAC-seq analysis
Cancer type-specific peak sets for all 23 TCGA ATAC-seq cancer types were acquired from ref. 29. The centers of the peaks were defined as the summits and for each cancer type the TE enrichment was calculated as described above. The organ system of origin annotations were acquired from ref. 104. and the tumor histology data were acquired from ref. 105.

## Roadmap chromatin state analysis
25-state chromatin state annotations for 127 Roadmap tissue types were acquired from ref. 27. GP5d STARR-seq peak summits overlapping MER11 elements were filtered and the enrichment analysis of the chromatin states in the summit loci was performed as described in the "TE enrichment analysis".

## ATAC-seq, ChIP-seq, TT-seq/GRO-seq read alignment to MER11B/LTR12C consensus sequences
For GP5d, ATAC-seq, TFAP2A ChIP-seq, and TT-seq data were first aligned to the reference genome. Unique reads mapped to 92 MER11B elements that overlapped with STARR-seq peaks were extracted and mapped to the MER11B consensus sequence to obtain compiled reads.

For HepG2, ATAC-seq, NFYA ChIP-seq, and GRO-seq data were first aligned to the reference genome. Unique reads mapped to 597 LTR12C elements that overlapped with HepG2 STARR-seq peaks were extracted and mapped to LTR12C consensus sequence to obtain compiled reads. Bwtools v.1.0 (bwtools aggregate) was used to plot compiled signal for ATAC-seq, ChIP-seq, and TT-seq/GRO-seq at LTR consensus sequences[106].

## DepMap TFAP2A dependency analysis
DepMap CRISPRi screening data for TFAP2A was downloaded for colorectal and liver cell lines. CRISPR Chronos score (TFAP2A gene effect) and TFAP2A expression (Log2 (TPM + 1)) for colorectal and liver cells were plotted by using GraphPad Prism 9[55].

## Motif analyses
All motif enrichments for each TE subfamily or STARR cluster were analyzed with AME from the MEME suite v.5.0.2 with shuffled sequences as the background (ame–control–shuffle--)[107]. JASPAR 2022 CORE non-redundant vertebrate motif annotations were used as the motif file input. Motif-clustering data was downloaded from ref. 108. Output from AME was read into R and the E-values were −log10-transformed. Each motif hit from AME analysis was assigned to a corresponding cluster. For each cluster, the minimum E-value from all individual motif hits was selected, the columns were scaled with the R scale function (center = F) and plotted.

## Cluster analysis
In GP5d, RPKM-normalized bigwig files for STARR-seq, ATAC-seq, ChIP-seq for H3K4me1, H3K4me3, H3K9me3, H3K27ac, H3K27me3, H3K36me3, p53 and 5-fluorouracil treated p53 and unnormalized CpG methylation bigwig from NaNoMe-seq were used to create a matrix file with deepTools v.3.5.0[87] with a region of 3 kb in both directions around

STARR-seq peak summits as the center reference point. (compute-Matrix reference-point−referencePoint center−missingDataAsZero -a 3000 -b 3000). The resulting deepTools matrix was loaded into R and columns were centered with the R scale function. R k-means function was then used to cluster the matrix with centers from two to nine and options nstart = 50, iter.max = 15. The resulting output was analyzed by elbow plotting and five clusters were determined to be the optimal number of clusters. Similar to GP5d, HepG2, and RPE1 STARR-seq peaks were clustered.

### Activity-by-contact model analysis
Candidate regions for the ABC model[37] were created using GP5d ATAC-seq data (see "ATAC-seq analysis" for processing details) with the makeCandidateRegions.py with the ENCODE blacklist as excluded regions, 250 bp peak extension and 150000 selected peaks (---regions_blocklist ENCFF356LFX.bed, --peakExtendFromSummit 250, --nStrongestPeaks 150000). Enhancer activity was quantified using the run.neighborhoods.py script with ATAC-seq and H3K27ac ChIP-seq (see "ChIP-seq analysis" for processing details) and the expression table of TPM counts from Salmon analysis (see "RNA sequencing") and GENCODE v36 gene annotations as inputs.

Contacts were predicted with the predict.py script with the output from the previous steps with threshold set as .02, hic-resolution set at 5000, powerlaw scaling and GP5d H3K27ac Hi-ChIP data set as the contact mapping input (see "Hi-ChIP analysis" for processing details) (--scale_hic_using_powerlaw, --threshold .02, --hic_resolution 5000).

### Statistical analysis and plots
All statistical and downstream analyses were performed in R v.4.1.2[109]. Profile plots were created from the bigwig files with the R package soGGi v.1.24.1 using the regionPlot function with the option normalize = T[110]. The signal was smoothed with rollmean from Zoo package v.1.8-10[111]. Genomic annotation for STARR-seq peaks was performed with ChIPseeker[112]. All plotting was performed in ggplot2 v.3.3.6 from the Tidyverse suite v.1.3.1[113] except for the motif enrichments that were plotted with ComplexHeatmap v.2.8.0[114] and enrichment heatmaps that were plotted with EnrichedHeatmap v.1.22.0[115], with code adapted from[47]. Correlation analyses between biological and technical replicates were performed by using multiBigwigSummary v.3.1.3 and plotted with plotCorrelation v.3.1.3[87]. Illustrations were created with BioRender.com. Heatmaps in Figs. 3 and 4 were plotted by using deepTools[87].

### Reporting summary
Further information on research design is available in the Nature Portfolio Reporting Summary linked to this article.

### Data availability
Data generated in this study has been deposited in the GEO database under accession "GSE221053". The publicly available data was accessed as follows: GP5d and HepG2 STARR-seq data used in this study are available in GEO database under accession "GSE180158"[17]. For GP5d, genomic and p53-null STARR-seq data from "GSM5454433"[17] and "GSM5454435"[17], and for HepG2 genomic STARR replicates 1 and 2 from "GSM5454437"[17] and "GSM5454438"[17], respectively. GP5d ChIP-seq data for H3K27ac ("GSM5454417"[17]), H3K9me3 ("GSM5454420"[17]), H3K27me3 ("GSM5454428"[17]), 5-FU treated mIgG ("GSM5454414"[17]), untreated p53 ("GSM5454412"[17]) and 5-FU treated p53 ("GSM5454413"[17]) were acquired from the same study. GP5d H3K4me1 ("GSM1240814"[71]) was obtained with the GEO accession "GSE51234"[71]. HepG2 ChIP-seq data for H3K27ac, H3K27me3, H3K36me3, H3K4me1, H3K4me3, H3K9me3, p53, and NFYA were downloaded from "ENCODE [https://www.encodeproject.org/]"[69] with fastq file accessions "ENCFF000BFD"[69], "ENCFF001FLQ"[69], "ENCFF001FMA"[69], "ENCFF000BEX"[69], "ENCFF901NZE"[69], "ENCFF000BFK"[69], "ENCFF257UIJ"[69], and "ENCFF081VHA"[69], respectively. Replicate 1 for HepG2 ATAC-seq was downloaded with accessions

"ENCFF664UPL"[69] and "ENCFF289UIB"[69] for read file 1 and 2, respectively. GP5d TT-seq data was downloaded from GEO database under accession code "GSM4610669"[30]. HepG2 GRO-seq raw data were downloaded from GEO database under accession code "GSM2428726"[31]. RPE1 ATAC-seq ("GSM5366618"[116]) and ChIP-seq data for H3K27ac ("GSM5345550"[116]), H3K27me3 ("GSM5345534"[116]), H3K36me3 ("GSM5345454"[116]), H3K4me1 ("GSM5345374"[116]), H3K4me3 ("GSM5345406"[116]), H3K9me3 ("GSM5345502"[116]), were acquired from "GSE175752"[116] and p53 ("GSM2677386"[117]) ChIP-seq was acquired from "GSE100292"[117]. WGBS ("GSM3394824"[118]) data for RPE1 was acquired from "GSE120140"[118]. NCBI genome annotation files for GRCh38 were downloaded from Illumina iGenomes [http://igenomes.illumina.com.s3-website-us-east-1.amazonaws.com/Homo_sapiens/NCBI/GRCh38/Homo_sapiens_NCBI_GRCh38.tar.gz]. A gene annotation GTF file was acquired from Gencode Release 36 for the reference chromosomes [https://ftp.ebi.ac.uk/pub/databases/gencode/Gencode_human/release_36/gencode.v36.annotation.gtf.gz]. A repeatMasker.txt (2021-09-03) file was downloaded from the UCSC table browser [https://genome.ucsc.edu/cgi-bin/hgTables]. MER11B and LTR12C consensus sequences were acquired from RepBase [https://www.girinst.org/repbase/update/index.html]. GRCh38 chromosome sizes file (2020-03-13) file was downloaded from UCSC [https://hgdownload-test.gi.ucsc.edu/goldenPath/hg38/bigZips/latest/]. Unified GRCh38 blacklist BED file ("ENCFF356LFX"[69], release 2020-05-05) was downloaded from "ENCODE [https://www.encodeproject.org/]". Transcription factor motifs were acquired from JASPAR 2022 CORE non-redundant vertebrate annotations [https://jaspar.genereg.net/download/data/2022/CORE]. The position weight matrices in MEME format were used for downstream motif analyses. Motif-clustering data was downloaded from [https://resources.altius.org/~jvierstra/projects/motif-clustering-v2.0beta/]. TCGA cancer type-specific ATAC-seq peak sets were acquired from [https://gdc.cancer.gov/about-data/publications/ATACseq-AWG]. Roadmap 25-state chromatin model bed files for 127 cell types were acquired from [https://egg2.wustl.edu/roadmap/data/byFileType/chromhmmSegmentations/ChmmModels/imputed12marks/jointModel/final/]. Source data are provided with this paper.

### Code availability
The R scripts used in the analysis are publicly available in https://github.com/Karttune/Karttunen_Patel_et_al.

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

## Acknowledgements

We thank HiLIFE research infrastructures including the Biomedicum Functional Genomics Unit (FuGU), the FIMM NGS Genomics laboratory, and the Flow Cytometry Unit at the University of Helsinki. We thank the Center for Scientific Computing (CSC), Finland, for the computational infrastructure, Professor Lauri Aaltonen's laboratory facilities for genomics work, Pinja Perkkiö for technical help, Justyna Kolakowska and Inga-Lill Åberg for nanopore sequencing, and Dr. Päivi Pihlajamaa for the advice on the CRISPR validation experiments and critical reading of the manuscript. This study was supported by grants from the Academy of Finland to B.S. (317807, 320114, 346065), J.X. (342594), L.A. (Finnish Center of Excellence Program 2018–2025, 312041, Academy Professor grants 319083, 320149), iCAN Digital Precision Cancer Medicine Flagship (320185), Cancer Foundation Finland sr to B.S., L.A., Sigrid Jusélius Foundation to B.S., and Jane and Aatos Erkko Foundation to B.S.

## Author contributions

B.S. conceived and supervised the study and performed the STARR-seq experiment and TFAP2A ChIP-seq. K.K. performed STARR-seq and genomics data analysis together with D.P. who performed the ChIP-seq, TT-seq, and GRO-seq analysis, CRISPR/Cas9 validation, and epigenetic inhibition experiments. J.X. performed the NaNOMe-seq and Hi-ChIP and L.F. performed the ATAC-seq. K.P. helped with the initial processing of NaNOMe-seq data with support from L.A. B.S., K.K., and D.P. wrote the manuscript with contributions from all authors.

## Competing interests

The authors declare no competing interests.
