## [Peer Review File · Nature Communications]

Transposable elements as tissue-specific enhancers in cancers of endodermal lineageReviewers' Comments:

Reviewer #1:

Remarks to the Author:

Karttunen et al. present a manuscript discussing the role of transposable elements as enhancers in two cancer cell lines, HepG2 and GP5d. Based on STARR-Seq and epigenetic data from the cell lines they investigate the enhancers overlapping with TEs. They describe 5 enhancer clusters, 4 of them with enrichment of different TE families. They also describe common and tissue-specific enhancers and investigate their functional role in the regulation of gene expression.

The paper is well-written, both the text and the flow are clear. The figures support the text and follow a logical order. Both the data and the codes are available, the data seems reproducible. The authors used a wide variety of data types to conduct a thorough functional analysis of the TE enhancers. The topic is important to understand the role of TEs, especially during tumorigenesis. The data will be an important resource for further studies as well.

I have no major criticism. Two minor points are:

1. The authors mention the evolutionary young and old TEs multiple times. However, apart from Figure 1d, no figure or data supports their points. In the text they reference the Extended Data Figure 3b, which, in my opinion, doesn't contain such information. This is an interesting point, the authors should illustrate it better.
2. The authors could elaborate more on the implications and the novelty of their findings. How do the identified regions look in patients of the two cancer types, for example in their methylation levels? Are the genes they identified to be regulated in a tissue-specific manner different between the cancer types?

Reviewer #2:

Remarks to the Author:

In this article, Karttunen and colleagues investigate the regulatory properties of transposable elements (TEs) in the human genome using data from global enhancer assays, called STARR-seq, in a couple of colon and liver-derived cancer cell lines (GP5d and HepG2). Based on their experiments, they conclude that TEs display important enhancer activity that can be associated to various TE subsets or families. As expected from previous work the LTRs-containing TEs are generally found more frequently and active as compared to the other classes. Using large genomic and epigenomic data sets in GP5d, the authors characterize 5 classes of enhancers, including conventional enhancers (high H3K27ac, low H3K4me3/1), promoter-like enhancers (high H3K27ac, high H3K4me3/1) and 3 classes with little active and more repressive marks. These latter 3 groups contain most TE enhancers while groups 4 and 5 tend to bind p53. Specific TE classes are defined specifically in each class to the exception of class 2 (promoters). Follow-up analyses also indicate specific TE enhancer usage depending on the cell line as well as some common and specific TF enrichment. Furthermore, methylated library usage allows to analyse methylation impact on TE enhancer activity, yielding apparently limited impact in general. Finally, prediction of gene-enhancer pairs suggests a link between active TE enhancer and gene-specific expression in a given cell line.

This study addresses interesting questions regarding the contribution of TE to regulatory sequence landscape and more specifically enhancers. It essentially makes use of a very large data set published recently by the authors and the group of J. Taipale (PMID: 35190730) while few additional experiments were added in this work. In general, the study is sound but would greatly benefit from additional data to extend the observations to other cancer cell types (at least one or two) to generalize the observations. As mentioned by the authors also, one caveat of the analysis is the repetitive nature of the TEs, which makes a large fraction of the data not easily interpretable. Other issues regarding this work are also mentioned below.

Below are suggestions for improvement of the manuscript.

1- The analyses performed in Figure 2 (clustering) should be performed also in HepG2 cells and in another endoderm-independent cancer cell line. The analyses relating to the specificity (Fig. 3 and 4) and 5 should also be extended to another cell line.

2- My understanding is that this article as well as the previous article (PMID: 35190730) is making use of a single ChIP-seq and ATAC-seq replicate for most data analyzed. This is not in my opinion a proper standard for large scale studies. Additional replicates should be performed. Standard QC for reproducibility of the new STARR-seq replicates should be displayed in supplementary materials.

3- The conclusions yielded from the in vitro-methylated STARR-seq analyses are not clear to me. The plot shown in Fig. 5a shows observed/expected ratio STARR-seq ratio in the 2 conditions (methylated vs non-methylated) and further analyzed for RNA variation (Fig. 4b), validating that $\frac{3}{4}$ are indeed differential in between the 2 cell lines. However, from this plot and somehow the text, it seems that only these 4 were found statistically significant from the global analysis. Is this correct? If so, while one can acknowledge that these 3 sequences are different in the 2 cell lines, the majority of peaks are not different and the global effect of methylation is modest if not absent. I guess this should be clarified as for the conclusion of this section.

4- Based on Fig. 2a, it seems that cluster 3 is the one that is most methylated on CpG in vivo. How do the TE-enhancer sequences that are methylated in vivo (based on MOME-seq) behave in the non-methylated vs in vitro-methylated STARR-seq assay?

5- The ChIP-seq data (enrichment) of ZTB33/BANP over THE1B and THE1C (Fig. 5e) is not convincing as compared to the local background. So, the authors could either perform their own BANP ChIP-seq or consider removing this data from the manuscript.

6- TT-seq and GRO-seq signals shown in panel 4d are close the background. I guess in general, this should be compared to control of average enrichment for 'classical' enhancer observed in cluster 1 from Figure 1.

7- Expression levels in between different cell lines is difficult to monitor; have the authors used spike-in strategies to compare the 2?

8- Given the problem of the multiple alignments for most TEs, and given that the authors only used uniquely mapped reads, it would be good to know the statistics of TE that are found in uniquely vs multiple alignment sequence tags (to be included in one of the supplementary tables) in STARR-seq experiments.

Minor remarks

9- While the data sets used in this study are described in the methods, it would be easier for the reader to have a table that recapitulates all experiments used for analysis, including each biological replicate, their GSE and the publication they originate from.

10- I would not call the nucleosome observed in Fig. 4g 'highly phased'. Highly phased nucleosome would better refer to that of CTCF, active TSSs or pioneer transcription factors. Further on, nucleosome-ATAC procedure is not ideal for precise nucleosome positioning assessment.

11- None of the figures indicate the sample size in the side-by-side comparisons ($n=x$). This should be added, especially given that the box is ticked in the report summary.

Reviewer #3:

Remarks to the Author:

In this manuscript, Karttunen et al. explore to which extent transposable elements (TEs) can act as enhancers in two cancer cell lines, using massively parallel reporter assay (MPRA) datasets, combined with epigenetic feature assessment. Their main finding is that specific TE families can act as cell-type-specific enhancers while others can act more broadly (some being under the control of p53). The originality of the work comes primarily from the use of MPRA datasets that reveal the potential of a sequence to work as enhancer independently of the chromatin context and thus independently from repressive mechanisms that may restrict their activity in most cellular context. Another strength of the

work is that overall, the analyses are well executed with a great attention to details, figures are clear, and the manuscript is well-written. Nevertheless, the new findings or concepts appear limited to my opinion as most of these families were already proposed to act as enhancer or promoter in various tissues or conditions, and candidate transcription factors already identified.

Summary:

Positives:

- An original approach to identify and study the role of TEs acting as enhancers
- A thorough and in-depth computational analysis of enhancer-TEs in 2 cell lines
- The demonstration of cell-type-specific enhancers derived from TEs

Negatives:

- Most of the identified TEs were already known to act as enhancer (or promoter)
- The biological significance of in silico predicted TE-enhancers-gene contacts (and more generally of the TE-enhancers on gene expression) has not been addressed.

Major points:

1. It is not always clear whether analyses shown in figures or extended figures were done on all STARR-seq peaks or a subset corresponding to a specific cluster. It is important to know for the reader as only those present in cluster 1 show features of functional enhancers, whereas the majority (in cluster 3) are more related to cryptic enhancers and may never actually act as enhancers.
2. It remains unknown whether in silico predicted enhancers plays a significant role in gene expression. Some wet-lab validation by CRISPR-mediated deletion of candidate enhancers followed by gene expression analysis would strengthen the presented data.
3. Work on the transactivating roles of LTR12 as enhancer and promoter and its regulation by TF such as NFY has been extensive. LTR12 is the name of HERV9 LTR (also known as ERV-9) and therefore some of the literature may have escaped the attention of the authors on this topic. The findings of the present manuscript should be discussed in the light of the previous findings (see for ex. Long et al. Genomics 1998, Pi et al. PNAS 2004, Yu et al JBC 2009, Krönung et al Oncotarget 2016, Hu et al. NAR 2017, Deniz et al. Nat Commun 2020, Yu et al. Cell Stem Cell 2022, Iouranova et al. Mob DNA 2022, etc). Same thing with MER11-HERVK11 and their associated KZFP (see for ex. Bi et al. JBC 1991, Pontis et al. Nat Commun 2022, Imbeault et al Nature 2017).

Minor points :

4. Some TE-derived enhancers are expected to be located in intronic sequences. Are those excluded from the analysis when analyzing nascent transcription data (TT-seq and GRO-seq)?
5. I find the RNA-seq signal shown in genome browser views odd as it seems to cover the entire gene body irrespective of the exons/introns (e.g. Fig 3c, 4f, 6c-d). What kind of RNA-seq libraries were used? It could be just a matter of figure resolution. Please clarify.
6. Code used to generate figures was provided in the reviewer files, but the 'code availability' section indicates that it will be made available upon request. I encourage the authors to share it on a publicly accessible server (e.g. zenodo, github, etc) and to include the code for extended figures, too.

Point-by-point response to reviewers of Karttunen & Patel et al.

We thank the reviewers for their constructive comments. Please find our point-by-point response to all specific criticisms on the following pages, with reviewers' comments in *italic*, our response to them in roman, and the changes made to the manuscript indicated in **bold**. Affected page and line(s) are also indicated.

Reviewer #1:

Karttunen et al. present a manuscript discussing the role of transposable elements as enhancers in two cancer cell lines, HepG2 and GP5d. Based on STARR-Seq and epigenetic data from the cell lines they investigate the enhancers overlapping with TEs. They describe 5 enhancer clusters, 4 of them with enrichment of different TE families. They also describe common and tissue-specific enhancers and investigate their functional role in the regulation of gene expression.

The paper is well-written, both the text and the flow are clear. The figures support the text and follow a logical order. Both the data and the codes are available, the data seems reproducible. The authors used a wide variety of data types to conduct a thorough functional analysis of the TE enhancers. The topic is important to understand the role of TEs, especially during tumorigenesis. The data will be an important resource for further studies as well.

We thank the reviewer for the supportive comments and the interest in our results.

I have no major criticism. Two minor points are:

1. The authors mention the evolutionary young and old TEs multiple times. However, apart from Figure 1d, no figure or data supports their points. In the text they reference the Extended Data Figure 3b, which, in my opinion, doesn't contain such information. This is an interesting point, the authors should illustrate it better.

We thank reviewer for pointing this out. We apologize that **Extended Data Fig. 3b** was incorrectly referred in the text (line 248 in the first submission). **We have now corrected the text on page 10, lines 268-270 to refer to Extended Data Fig. 5b as intended.**

As suggested by the reviewer, we have now illustrated the point about evolutionary old and young TEs better by **i) adding a new figure panel (Extended Data Fig. 1c, referred in the text on page 4, lines 87-89)** showing enrichment of TEs with different evolutionary origins within active enhancers from three distinct cell lines and **ii) by clarifying Extended Data Fig. 5b. Moreover, we have added an additional reference to the Discussion section** (Andrews et al., 2023, PMID: 37104580, **page 25, lines 593-595**), describing our findings in the light of recently reported regulatory activities of young and old TEs.

2. The authors could elaborate more on the implications and the novelty of their findings. How do the identified regions look in patients of the two cancer types, for example in their methylation levels? Are the genes they identified to be regulated in a tissue-specific manner different between the cancer types?

We thank the reviewer for the valuable suggestion to extend the analysis of TE activity in the context of patient data. For this, **we have added new analysis utilizing ATAC-seq data from 23 cancer types from The Cancer Genome Atlas (TCGA) datasets to study whether the TEs identified in cell lines are also enriched within open chromatin regions in patient data.** Based on this analysis, 11 out of the 18 GP5d-specific subfamilies were also enriched in TCGA colorectal adenocarcinoma data, and in general, TE enrichment clustered remarkably well together with the cancer organ system of origin, suggesting cancer type-specific TE activation and supporting our findings from the cancer cell lines. **This analysis is added to new Extended Data Fig. 6 and discussed in pages 11-12, lines 298-310.**

We also looked into the methylation analysis suggested by the reviewer. However, whole-genome methylation data that covers the non-coding regulatory regions including the repetitive regions is not easily available. Most of the patient methylation data available from TCGA is from the Illumina 450k array, where only a small minority of the probes cover the TEs that we have focused on in this manuscript, highlighting the need for a more comprehensive approach that also includes the non-coding and repetitive regions in the analysis of methylation and patient data in the future.

Reviewer #2

In this article, Karttunen and colleagues investigate the regulatory properties of transposable elements (TEs) in the human genome using data from global enhancer assays, called STARR-seq, in a couple of colon and liver-derived cancer cell lines (GP5d and HepG2). Based on their experiments, they conclude that TEs display important enhancer activity that can be associated to various TE subsets or families. As expected from previous work the LTRs-containing TEs are generally found more frequently and active as compared to the other classes. Using large genomic and epigenomic data sets in GP5d, the authors characterize 5 classes of enhancers, including conventional enhancers (high H3K27ac, low H3K4me3/1), promoter-like enhancers (high H3K27ac, high H3K4me3/1) and 3 classes with little active and more repressive marks. These latter 3 groups contain most TE enhancers while groups 4 and 5 tend to bind p53. Specific TE classes are defined specifically in each class to the exception of class 2 (promoters). Follow-up analyses also indicate specific TE enhancer usage depending on the cell line as well as some common and specific TF enrichment. Furthermore, methylated library usage allows to analyse methylation impact on TE enhancer activity, yielding apparently limited impact in general. Finally, prediction of gene-enhancer pairs suggests a link between active TE enhancer and gene-specific expression in a given cell line.

This study addresses interesting questions regarding the contribution of TE to regulatory sequence landscape and more specifically enhancers. It essentially makes use of a very large data set published recently by the authors and the group of J. Taipale (PMID: 35190730) while few additional experiments were added in this work. In general, the study is sound but would greatly benefit from additional data to extend the observations to other cancer cell types (at least one or two) to generalize the observations. As mentioned by the authors also, one caveat of the analysis is the repetitive nature of the TEs, which makes a large fraction of the data not easily interpretable. Other issues regarding this work are also mentioned below.

We thank the reviewer for the interest and excellent constructive comments.

Below are suggestions for improvement of the manuscript.

1- The analyses performed in Figure 2 (clustering) should be performed also in HepG2 cells and in another endoderm-independent cancer cell line. The analyses relating to the specificity (Fig. 3 and 4) and 5 should also be extended to another cell line.

We thank the reviewer for the suggestion and agree that adding a non-endodermal cell line strengthens the comparative analysis and overall conclusions of the manuscript. To address this point, **we performed STARR-seq experiment from a new cell line, a retinal pigment epithelial cell line RPE1 that is of ectodermal origin, introduced on page 4, lines 75-77 of the revised manuscript. General TE enrichment and overlap analysis of RPE1 STARR-seq data together with GP5d and HepG2 data is shown in new Extended Data Fig. 1 and described on page 4, lines 77-84.** As suggested by the reviewer, **we performed the clustering analysis along with cluster-level TE enrichment and TF motif analyses also for HepG2 and RPE1 (shown in new Extended Data Fig. 3 and described on page 6-7, lines 147-161 and page 7, lines 171-181).** Overall, all three cell types show enrichment of the LTR class TEs, but have significant differences on the activity of distinct TE subfamilies.

2- My understanding is that this article as well as the previous article (PMID: 35190730) is making use of a single ChIP-seq and ATAC-seq replicate for most data analyzed. This is not in my opinion a proper standard for large scale studies. Additional replicates should be performed. Standard QC for reproducibility of the new STARR-seq replicates should be displayed in supplementary materials.

We understand the concern and thank the reviewer for pointing this out. **We have now performed replicate ChIP-seq experiments for TFAP2A and H3K36me3 and included the Pearson correlation data and representative IGV snapshots in Extended Data Fig. 12 (described in text page 33, line 849-851).** Most of the used STARR-seq data has been previously published and the QC is available in PMID: 35190730. **Here, we performed correlation analysis and included Pearson correlation data for the previously new STARR-seq datasets, i.e. HepG2 with methylated library and RPE1 (Extended Data Fig. 12, described in text page 32, line 833).** We have also included Pearson correlation analysis results for GP5d ATAC-seq replicates (replicate 1 from this study and replicate 2 from PMID: 35190730) in Extended Data Fig 12, described in text page 33, line 845.

3- The conclusions yielded from the *in vitro*-methylated STARR-seq analyses are not clear to me. The plot shown in Fig. 5a shows observed/expected ratio STARR-seq ratio in the 2 conditions (methylated vs non-methylated) and further analyzed for RNA variation (Fig. 4b), validating that ¼ are indeed differential in between the 2 cell lines. However, from this plot and somehow the text, it seems that only these 4 were found statistically significant from the global analysis. Is this correct? If so, while one can acknowledge that these 3 sequences are different in the 2 cell lines, the majority of peaks are not different and the global effect of methylation is modest if not absent. I guess this should be clarified as for the conclusion of this section.

We thank the reviewer for the comment and apologize for the unclear legend for Figure 5. The reviewer is correct in that only four subfamilies were significantly different in their enrichment between the non-methylated and methylated STARR-seq libraries in HepG2 (**Fig. 5a**). These TEs were selected for further analysis in the subsequent figures; most of the analysis in Figure 5 is between methylated and non-methylated libraries in HepG2 cells and not a comparison between two cell types, which is only the case in **Fig. 5f**. **We have now clarified the legend for Fig. 5a and 5b to make it clearer that this is comparison between two STARR-seq libraries in HepG2 and that the comparison in Fig. 5b is between the subfamilies that were found significant in Fig. 5a (page 20, lines 458-464).**

We have also added a clarifying figure to this rebuttal document (**Rebuttal Figure 1**), showing that STARR-seq signal is consistently reduced upon library methylation (discussed in more detail in the response to point 4.) The caveat of this analysis is that STARR-seq is episomal, leading to relatively minor differences seen in methylated vs. non-methylated STARR-seq. As suggested by reviewer, **we have clarified the relatively modest effect of DNA methylation on STARR-seq activity while concluding this section (page 19, line 452-453).**

4- Based on Fig. 2a, it seems that cluster 3 is the one that is most methylated on CpG *in vivo*. How do the TE-enhancer sequences that are methylated *in vivo* (based on MOME-seq) behave in the non-methylated vs *in vitro*-methylated STARR-seq assay?

We thank the reviewer for this interesting question. We have looked at the data in more detail and added a clarifying figure below, showing that the STARR-seq signal in GP5d non-methylated (NM) vs. methylated (M) libraries at TEs is consistently reduced between different clusters (shown for MER11A as a representative example in the **Rebuttal Figure 1, top panel**). However, it seems that evolutionarily young subfamilies, such as CG-rich LINE elements (L1PA2, L1PA3) as well as MER11 elements show a more drastic loss of signal, whereas for example MER61E shows only a minor reduction in signal (**Rebuttal Figure 1, bottom panel**). Due to difficulties in interpreting these results, we decided omit this analysis from the manuscript.

Rebuttal Figure 1: Comparison of STARR-seq signal in GP5d cells from non-methylated (NM) vs. methylated (M) libraries. **Top panel:** Representative figure of the loss in STARR-seq signal between MER11A elements in GP5d clusters (Fig. 2a in the manuscript), showing a consistent loss of the signal in every cluster. **Bottom panel:** Percentage gain/loss (GP5d M / NM) in STARR-seq signal in representative TEs in GP5d, showing a higher loss of signal in some young, CG-rich TEs, e.g. L1HS, L1PA2, and L1PA3.

5- The ChIP-seq data (enrichment) of ZTB33/BANP over THE1B and THE1C (Fig. 5e) is not convincing as compared to the local background. So, the authors could either perform their own BANP ChIP-seq or consider removing this data from the manuscript.

We thank the reviewer for this important comment. To address this, we performed ZBTB33 ChIP-seq in HepG2 cells as suggested by the reviewer and did not observe strong enrichment at THE1 elements. Thus, we decided to remove the ENCODE ZBTB33 ChIP-seq data from Fig. 5e, as suggested by the reviewer.

6- TT-seq and GRO-seq signals shown in panel 4d are close the background. I guess in general, this should be compared to control of average enrichment for 'classical' enhancer observed in cluster 1 from Figure 1.

We thank reviewer for pointing this out. We have now plotted TT-seq and GRO-seq signals for classical enhancers from cluster 1 from GP5d and HepG2 cells as suggested by the reviewer. These new plots are shown in **Extended Data Fig 7c-d**, respectively. However, we observed stronger nascent transcriptional activity from MER11B elements in GP5d and LTR12C in HepG2 as compared to cluster 1 enhancers of respective cell types (**Fig. 4d, Extended Data Fig 7c-d**). **These comparison results are now addressed on page 15, lines 356-358.**

7- Expression levels in between different cell lines is difficult to monitor; have the authors used spike-in strategies to compare the 2?

We thank the reviewer for this comment. We are planning to use the spike-in strategies in future experiments, but in the experiments from this manuscript, spike-in strategies were not used. **This is now clarified in the Methods section (page 30, line 742).**

8- Given the problem of the multiple alignments for most TEs, and given that the authors only used uniquely mapped reads, it would be good to know the statistics of TE that are found in uniquely vs multiple alignment sequence tags (to be included in one of the supplementary tables) in STARR-seq experiments.

We thank the reviewer for this comment. **We have now added read mapping statistics to Supplementary Table 7 (referred to on manuscript page 32, lines 831-832).** To avoid false positives, we have only used the uniquely mapping reads in our analysis, and we are aware that this may lead to an underestimation of actual levels of TE enrichment. However, using methods of assigning multimapping reads to sequences are not without problems, which is why we decided to use only uniquely mapping reads for this first-of-its-kind MPRA measurements for studying TEs. **These considerations have now been clarified in the Discussion section (page 28, lines 680-681) and Methods section (page 32, lines 819-821).**

9- While the data sets used in this study are described in the methods, it would be easier for the reader to have a table that recapitulates all experiments used for analysis, including each biological replicate, their GSE and the publication they originate from.

We thank the reviewer for the valuable suggestion. **We have now included a new Supplementary table 8 and given details of all the used data in it,** as suggested by the reviewer.

10- I would not call the nucleosome observed in Fig. 4g 'highly phased'. Highly phased nucleosome would better refer to that of CTCF, active TSSs or pioneer transcription factors. Further on, nucleo-ATAC procedure is not ideal for precise nucleosome positioning assessment.

We thank the reviewer for pointing this out. We agree that nucleo-ATAC procedure is not ideal for nucleosome positioning assessment. Thus, we have decided to remove figure panels **Fig. 4g** and **Extended Data Fig. 8a (figure numbers as in the first submission)** describing the nucleo-ATAC-predicted nucleosomal positioning around TFAP2 binding sites, since they do not affect the overall conclusions of the manuscript.

11- None of the figures indicate the sample size in the side-by-side comparisons (n=x). This should be added, especially given that the box is ticked in the report summary.

We thank the reviewer for the comment and apologize for this omission. The sample sizes have been added to all relevant figures and/or figure legends: **Fig. 4g (page 18, line 420), 5b (page 20) and 5f (page 20); Extended Data Fig. 4c (page 7) and 5b (page 8).**

Reviewer #3:

In this manuscript, Karttunen et al. explore to which extent transposable elements (TEs) can act as enhancers in two cancer cell lines, using massively parallel reporter assay (MPRA) datasets, combined with epigenetic feature assessment. Their main finding is that specific TE families can act as cell-type-specific enhancers while others can act more broadly (some being under the control of p53). The originality of the work comes primarily from the use of MPRA datasets that reveal the potential of a sequence to work as enhancer independently of the chromatin context and thus independently from repressive mechanisms that may restrict their activity in most cellular context. Another strength of the work is that overall, the analyses are well executed with a great attention to details, figures are clear, and the manuscript is well-written. Nevertheless, the new findings or concepts appear limited to my opinion as most of these families were already proposed to act as enhancer or promoter in various tissues or conditions, and candidate transcription factors already identified.

Summary

Positives:

- *An original approach to identify and study the role of TEs acting as enhancers*
- *A thorough and in-depth computational analysis of enhancer-TEs in 2 cell lines*
- *The demonstration of cell-type-specific enhancers derived from TEs*

Negatives:

- *Most of the identified TEs were already known to act as enhancer (or promoter)*
- *The biological significance of in silico predicted TE-enhancers-gene contacts (and more generally of the TE-enhancers on gene expression) has not been addressed.*

We thank the reviewer for the comments and appreciate the similar view on rationale of using STARR-seq for studying TEs. We agree that most of these elements have been reported in different contexts as enhancers but have not been analyzed in a comprehensive manner, and the activation of these elements remain understudied in cancers. As the reviewer noted in the positives, we have used an unbiased genome-wide analysis, a massively parallel reporter assay, to show how these elements can be exploited by cell-type specific TFs, paving the way for future detailed analysis of these difficult TEs. Importantly, we have now also experimentally validated the function of the identified TE enhancers in regulating mRNA expression of predicted target genes, as suggested by the reviewer (detailed response to point 2 below). We feel that this has substantially strengthened the novelty of our manuscript and we thank the reviewer for this valuable suggestion. We also added novel results at **Extended Data Fig. 5c** that strongly suggest that the MER11 elements previously identified as enhancers in embryonic tissues (PMID: 36418324) are reactivated in cancers, highlighting the role of dedifferentiation of cancer cells for the activation of these sequences, further increasing the novelty of our findings.

Major points:

1. It is not always clear whether analyses shown in figures or extended figures were done on all STARR-seq peaks or a subset corresponding to a specific cluster. It is important to know for the reader as only those present in cluster 1 show features of functional enhancers, whereas the majority (in cluster 3) are more related to cryptic enhancers and may never actually act as enhancers.

We thank the reviewer for pointing this out and apologize for the omission. We have now clarified this in the main text and in all relevant figure legends: **Fig. 1c-d (page 5, lines 98 and 102)**, **2a-d (page 9, line 195, 200, 204 and 212)**, **3a (page 14, line 313)**, **5a (page 20, line 458)**; **Extended Data Fig. 7c-d (page 11)** and **9a (page 15)**.

2. It remains unknown whether in silico predicted enhancers plays a significant role in gene expression. Some wet-lab validation by CRISPR-mediated deletion of candidate enhancers followed by gene expression analysis would strengthen the presented data.

We thank the reviewer for this important comment. As suggested, **we have now performed validation experiments using CRISPR-mediated deletion of specific TE sequences, followed by gene expression analysis by qRT-PCR. These results are shown in new panels Fig. 6c-d and Extended Data Fig. 10-11 and described under the subheading “MER11-derived STARR-seq enhancers transactivate nearby genes in GP5d cells” in the Results section (page 22, lines 527-542).** Importantly, we show that deletion of several identified TE enhancers affect the expression of their target genes, and we feel that these new results have greatly increased the biological significance of our manuscript.

3. Work on the transactivating roles of LTR12 as enhancer and promoter and its regulation by TF such as NFY has been extensive. LTR12 is the name of HERV9 LTR (also known as ERV-9) and therefore some of the literature may have escaped the attention of the authors on this topic. The findings of the present manuscript should be discussed in the light of the previous findings (see for ex. Long et al. Genomics 1998, Pi et al. PNAS 2004, Yu et al JBC 2009, Krönung et al Oncotarget 2016, Hu et al. NAR 2017, Deniz et al. Nat Commun 2020, Yu et al. Cell Stem Cell 2022, Iouranova et al. Mob DNA 2022, etc). Same thing with MER11-HERVK11 and their associated KZFP (see for ex. Bi et al. JBC 1991, Pontis et al. Nat Commun 2022, Imbeault et al Nature 2017).

We thank the reviewer for pointing this out and apologize for missing to cite these previous important results. **We have now taken into account and cited the previous literature. The manuscript text has been modified in the Results and Discussion sections (page 10-11, lines 272-273, page 25, lines 582-584 and page 27, 658-662).** Related to this, **we have also included new analysis showing that the TE copies that are active in GP5d cells reside in open chromatin regions exclusively in embryonic stem cells (ESC), induced ESCs and induced pluripotent stem cells in Roadmap tissues,** suggesting that cancers may utilize the same TEs that have been exapted as enhancers in embryonic development (**new Extended Data Fig. 5c and corresponding text on page 10-11, lines 272-279**).

Minor points :

4. Some TE-derived enhancers are expected to be located in intronic sequences. Are those excluded from the analysis when analyzing nascent transcription data (TT-seq and GRO-seq)?

We thank the reviewer for the comment. TE enhancers located in the intronic regions have not been excluded from the analysis, as intronic TEs can also serve as enhancers to upregulate nearby genes. **This has now been clarified in the Methods section (page 34, line 888-889).**

5. I find the RNA-seq signal shown in genome browser views odd as it seems to cover the entire gene body irrespective of the exons/introns (e.g. Fig 3c, 4f, 6c-d). What kind of RNA-seq libraries were used? It could be just a matter of figure resolution. Please clarify

We thank the reviewer for the important comment and apologize for the confusion. **We have now modified Figures 2c, 4e, 6a, 6b to represent the RNA-seq data better. We have also clarified the methodological part, stating that Stranded RNA-seq libraries were prepared by using KAPA stranded mRNA-seq kit (Roche), in the Methods section (page 31, line 786-788).**

6. Code used to generate figures was provided in the reviewer files, but the ‘code availability’ section indicates that it will be made available upon request. I encourage the authors to share it on a publicly accessible server (e.g. zenodo, github, etc) and to include the code for extended figures, too.

We apologize for the confusion. We actually had provided all the files at Editor’s request for the reviewer access after the initial submission, but the manuscript text was not updated at that point. As suggested by the reviewer, **the code for reproducing all the figures is now deposited to a GitHub repository that will be made available upon publication, and the link is included in the text (page 38, lines 1037-1038).**

Reviewers' Comments:

Reviewer #1:

Remarks to the Author:

I am satisfied with the response the authors gave and recommend the manuscript for publication.

Reviewer #2:

Remarks to the Author:

The authors have appropriately addressed most of my comments and the manuscript has improved. However, there is one minor point that remains problematic that is the quality of the replicates shown in Extended data Fig. 12, where the signal to noise is extremely variable in between experiments. While rep1 vs rep2 differences are acceptable for ATAC-seq and TFAP2A, rep2 from H3K36me3 is certainly not. This data set shows signal everywhere in intergenic regions and no clear enrichment in coding regions which should be the hallmark of this histone PTM. Thus I recommend either to reproduce this data set or to remove H3K36me3 analysis since it does not represent a major point in the manuscript.

Reviewer #3:

Remarks to the Author:

The authors have addressed most of my previous concerns. In particular, they now provide experimental validation for the role of some transposable elements as enhancers through Cas9-mediated deletion, as well as new MPRA data for a non-cancerous cell type. This clearly brings further value to the work. Therefore, I am happy to recommend publication once the minor points below have been resolved.

Minor points:

1. References to previous literature have been improved, but several seminal studies on LTR12 are still not cited (stressed in my initial review – see point #3 in there).

2. When describing cluster 4 (sequences with high MPRA activity on plasmids and high levels of the repressive mark H3K9me3 in a chromosomal context), the authors might wish to cite Zhu Y et al. Mol Cell 2012 (PMID: 22633489), as this article seems to describe similar features.

3. I was unable to access the code at any stage of the reviewing process, either through the submission system or through the provided GitHub link (broken at this time).

Point-by-point response to reviewers of Karttunen & Patel et al.

We thank the reviewers for their comments on our revised manuscript. Please find our point-by-point response to all specific criticisms on the following pages, with reviewers' comments in *italic*, our response to them in roman, and the changes made to the manuscript indicated in **bold**. Affected page and line(s) are also indicated.

Reviewer #1:

I am satisfied with the response the authors gave and recommend the manuscript for publication.

We thank the reviewer for reviewing our manuscript and recommending it for publication.

Reviewer #2:

The authors have appropriately addressed most of my comments and the manuscript has improved.

We thank reviewer and agree that the suggested revisions have significantly improved the manuscript.

However, there is one minor point that remains problematic that is the quality of the replicates shown in Extended data Fig. 12, where the signal to noise is extremely variable in between experiments. While rep1 vs rep2 differences are acceptable for ATAC-seq and TFAP2A, rep2 from H3K36me3 is certainly not. This data set shows signal everywhere in intergenic regions and no clear enrichment in coding regions which should be the hallmark of this histone PTM. Thus I recommend either to reproduce this data set or to remove H3K36me3 analysis since it does not represent a major point in the manuscript.

We thank the reviewer for pointing this out. We have removed the previous H3K36me3 ChIP-seq rep2 data and added a new H3K36me3 ChIP-seq rep2, which shows better correlation with rep1 (updated **Extended Data Fig. 12a**). Now both H3K36me3 ChIP-seq replicates show enrichment at coding regions (**Extended Data Fig. 12b-c**). The H3K36me3 ChIP-seq rep2 file has also been replaced in the GEO submission and detailed in the Nature reporting summary file.

Reviewer #3:

The authors have addressed most of my previous concerns. In particular, they now provide experimental validation for the role of some transposable elements as enhancers through Cas9-mediated deletion, as well as new MPRA data for a non-cancerous cell type. This clearly brings further value to the work. Therefore, I am happy to recommend publication once the minor points below have been resolved.

We thank reviewer and agree that the suggested revisions have significantly improved the manuscript.

Minor points:

1. References to previous literature have been improved, but several seminal studies on LTR12 are still not cited (stressed in my initial review – see point #3 in there).

We thank reviewer for pointing this out. We have now included the remaining suggested references (Long et al. Genomics 1998, Yu et al. Cell Stem Cell 2022, Krönung et al Oncotarget 2016) and the manuscript text has been modified in the Discussion section (page 16, lines 505-506, page 16, lines 511-513). Of note, one of the suggested references (Iouranova et al. Mob DNA 2022), was already cited in our first submission.

Reviewer also suggested studies Bi et al. JBC 1991 and Yu et al JBC 2009, which we regrettably were unable to find. However, in the case of Yu et al JBC 2009, reviewer might refer to Yu et al JBC 2005 (PMID:16105833), describing NF-Y mediated regulation of enhancer activity of ERV9 elements, which

we have now cited in the Discussion section (page 16, lines 505-506). In conclusion, we wish to thank the reviewer for their suggestions that helped us to comprehensively cite previous literature in the field.

2. *When describing cluster 4 (sequences with high MPRA activity on plasmids and high levels of the repressive mark H3K9me3 in a chromosomal context), the authors might wish to cite Zhu Y et al. Mol Cell 2012 (PMID: 22633489), as this article seems to describe similar features.*

We have included the suggested reference in the manuscript text (page 15, lines 489-491).

3. *I was unable to access the code at any stage of the reviewing process, either through the submission system or through the provided GitHub link (broken at this time).*

We apologize for the inconvenience in accessing the custom code through the GitHub link. In fact, we had provided the code to the editor for the reviewer access in the initial submission, and at least reviewer #1 had been able to access it as they stated that *“Both the data and the codes are available, the data seems reproducible”*. We have now made the GitHub repository publicly available, and the link is included in the text (page 29, lines 936-937).